# Heritability of the HIV-1 reservoir size and decay under long-term suppressive ART

Chenjie Wan [1,2,34], Nadine Bachmann [1,3,34], Venelin Mitov [4,5], François Blanquart[6], Susana Posada Céspedes [5,7], Teja Turk [1,3], Kathrin Neumann [1,3], Niko Beerenwinkel [7], Jasmina Bogojeska[8], Jacques Fellay [9,10], Volker Roth [11], Jürg Böni [3], Matthieu Perreau[12], Thomas Klimkait [13], Sabine Yerly [14], Manuel Battegay[15], Laura Walti[16], Alexandra Calmy[17], Pietro Vernazza[18], Enos Bernasconi [19], Matthias Cavassini[20], Karin J. Metzner [1,3,34], Huldrych F. Günthard [1,3,34], Roger D. Kouyos [1,3,34✉] & the Swiss HIV Cohort Study*

The HIV-1 reservoir is the major hurdle to curing HIV-1. However, the impact of the viral genome on the HIV-1 reservoir, i.e. its heritability, remains unknown. We investigate the heritability of the HIV-1 reservoir size and its long-term decay by analyzing the distribution of those traits on viral phylogenies from both partial-*pol* and viral near full-length genome sequences. We use a unique nationwide cohort of 610 well-characterized HIV-1 subtype-B infected individuals on suppressive ART for a median of 5.4 years. We find that a moderate but significant fraction of the HIV-1 reservoir size 1.5 years after the initiation of ART is explained by genetic factors. At the same time, we find more tentative evidence for the heritability of the long-term HIV-1 reservoir decay. Our findings indicate that viral genetic factors contribute to the HIV-1 reservoir size and hence the infecting HIV-1 strain may affect individual patients' hurdle towards a cure.

[1] Department of Infectious Diseases and Hospital Epidemiology, University Hospital Zurich, Zurich, Switzerland. [2] Department of Environmental Systems Science, ETH Zurich, Zurich, Switzerland. [3] Institute of Medical Virology, University of Zurich, Zurich, Switzerland. [4] Department of Biosystems Science and Engineering, ETH Zurich, Zurich, Switzerland. [5] Swiss Institute of Bioinformatics, Lausanne, Switzerland. [6] Center for Interdisciplinary Research in Biology, Collège de France, Paris, France. [7] Department of Biosystems Science and Engineering, ETH Zurich, Basel, Switzerland. [8] IBM Research - Zurich, Zurich, Switzerland. [9] School of Life Sciences, EPFL, Lausanne, Switzerland. [10] Precision Medicine Unit, Lausanne University Hospital and University of Lausanne, Lausanne, Switzerland. [11] Department of Mathematics and Computer Science, University of Basel, Basel, Switzerland. [12] Division of Immunology and Allergy, Centre Hospitalier Universitaire Vaudois, University of Lausanne, Lausanne, Switzerland. [13] Molecular Virology, Department Biomedicine - Petersplatz, University of Basel, Basel, Switzerland. [14] Laboratory of Virology, Geneva University Hospital, Geneva, Switzerland. [15] Department of Infectious Diseases and Hospital Epidemiology, University Hospital Basel, University of Basel, Basel, Switzerland. [16] Department of Infectious Diseases, Inselspital, Bern University Hospital, University of Bern, Bern, Switzerland. [17] Institute of Global Health, University of Geneva, Geneva, Switzerland. [18] Division of Infectious Diseases and Hospital Epidemiology, Cantonal Hospital of St. Gallen, St. Gallen, Switzerland. [19] Infectious Diseases Service, Regional Hospital, Lugano, Switzerland. [20] Division of Infectious Diseases, Centre Hospitalier Universitaire Vaudois, University of Lausanne, Lausanne, Switzerland. [34] These authors contributed equally: Chenjie Wan, Nadine Bachmann, Karin J. Metzner, Huldrych F. Günthard, Roger D. Kouyos. *A list of authors and their affiliations appears at the end of the paper. ✉email: Roger.Kouyos@usz.ch

Combination antiretroviral therapy (ART) can effectively block HIV-1 replication and reduce plasma virus levels to below the detection limit of clinical assays[1]. However, treatment cannot eradicate HIV-1 due to the existence of the extremely slowly decaying HIV-1 reservoir[2,3]. The HIV-1 reservoir refers to the proviral HIV-1 DNA that persists mainly in infected resting memory CD4+ T cells throughout the body[4], including the brain, lymph nodes, blood, and digestive tract. It is established already early during primary infection and persists even in patients under long-term ART with no detectable viremia[2,5–7]. Thus, the HIV-1 reservoir is recognized as a major hurdle to complete viral eradication.

As individuals with smaller HIV-1 reservoirs should be more amenable to cure, several studies have assessed the potential clinical, immunological, and epidemiological determinants of HIV-1 reservoir size. Early ART initiation limits the HIV-1 reservoir size[8–10]. Also, pre-treatment viral load correlates positively with reservoir size[3,11]. Immunological factors such as homeostatic proliferation[12], clonal expansion[13–15], and initial antiviral immune responses[16,17] and epidemiological factors such as transmission group and ethnicity[3] also play a role. The decay of the HIV-1 reservoir in individuals on ART and its potential determinants were only examined in a few studies[2,3,18–22]. Recently, the frequency of blips has also been associated with the size and a slower decay of the HIV-1 reservoir[3] and whether cryptic replication plays a role in refilling the reservoir has been an ongoing debate for decades[4]. Generally, a small to zero decay slope was reported, and viral blips were confirmed to slow down the decay of the latent HIV-1 reservoir[2,3]. However, the relative extent to which the reservoir size and decay are controlled by viral genetics still remains unknown.

The impact of viral genetic factors can be quantified as the viral heritability, which is defined as the fraction of total phenotypic variance explained by genetic factors. It ranges from 0% when genetic factors do not contribute to the phenotypic variance, to 100% when genetic factors explain the entire phenotypic variance[23]. The estimation of heritability depends on the partitioning of the observed variance into contributions from environmental factors and genetic factors. Heritability can be estimated using resemblance estimators, which measure the relative trait-similarity within transmission clusters. Established methods include parent–offspring (PO) regression[24–28] and analysis of variance with mixed-effect models[29,30]. Additionally, phylogenetic comparative methods can also be used to estimate heritability by measuring the association between observed trait values from individuals and their transmission tree inferred from pathogen sequences[24,25]. Common approaches include phylogenetic mixed models with an underlying Brownian motion process (PMM)[31–34] and phylogenetic mixed models with an underlying Ornstein Uhlenbeck process (POUMM)[28,30,35–37].

Most of these methods have been applied to estimate the heritability of HIV-1 related phenotypes. Among these, the influence of viral genetics on the SPVL has been the main focus of attention (see[28] and references therein). Generally, the consensus has been achieved that SPVL is heritable, with heritability estimates of 20–30% in different populations. Studies investigating the heritability of the CD4+ T cell decline, which is the most relevant measure of progression to AIDS, reported relatively low heritability estimates of 10–20%[36,37]. The heritability of the antibody response induced by an HIV infection has been estimated to be around 10–15% based on mixed-effect Tobit models[38].

In this study, we investigate the heritability of the HIV-1 reservoir size and long-term decay using viral sequences and total HIV-1 DNA measurements from Swiss HIV Cohort Study (SHCS) participants infected with HIV-1 subtype B. Total HIV-1 DNA was found to be a sensitive, clinically relevant HIV-reservoir marker which can be determined for large patient populations[3,19,39,40]. Both partial *pol* Sanger sequences, which were obtained for genotypic resistance testing, and viral near full-length genome sequences are considered to increase the resolution of our estimates. To avoid potential bias introduced by a single model, we explore both non-parametric (mixed-effect model) and parametric (phylogenetic mixed model assuming a trait evolution according to the BM and OU process) models. Further, we adjust our analysis for a broad range of viral and host characteristics known to influence the size and the decay of the HIV-1 reservoir.

## Results

**Study population**. We assessed the heritability of the HIV-1 reservoir size and decay slope in people living with HIV under long-term suppressive combination ART. HIV-1 reservoir was measured using total HIV-1 DNA measurements, a sensitive marker for the HIV-1 reservoir. The analysis was performed on the basis of the transmission network of 610 well-characterized patients infected with HIV-1 subtype B enrolled in the SHCS (Table 1).

From 1057 individuals enrolled in the SHCS with successfully quantified total HIV-1 DNA at least ~1.5, ~3.5, and ~5.4 years after the initiation of ART, we identified 475 individuals with available next generation sequences (NGS) of viral near full-length genome (denoted as population $A_0$) and 869 individuals with available Sanger sequences of partial *pol* region obtained for genotypic resistance testing (GRT) (denoted as population $B_0$) (Fig. 1).

Considering the inter-subtype heterogeneity of HIV-1, we restricted our study populations to individuals infected with HIV-1 subtype B strains, which were 351 individuals with available NGS sequences (denoted as population A) and 610 individuals with available Sanger sequences (denoted as population B) (Fig. 1). In the larger population (population B), 532 (87.2%) were male and 403 (66.1%) were men who have sex with men (MSM). 348 (99.1%) patients in the smaller population A also belonged to population B and population A had comparable characteristics to population B (Table 1). We additionally performed sensitivity analyses on the full datasets (population $A_0$ and $B_0$) including all HIV-1 subtypes (Supplementary Table 1).

To estimate the heritability using mixed-effect models, we extracted transmission clusters from phylogenies such that the maximum phylogenetic distance within the cluster was not larger than the pre-defined threshold. The number of extracted transmission clusters varied with different thresholds and different types of sequences used for phylogenetic inference (Supplementary Table 2). In particular, applying phylogenetic distance threshold of 0.04, 0.05, 0.06, and 0.09 substitutions per site for sequences obtained by NGS and comparable distance threshold levels of 0.01, 0.02, 0.03, and 0.045 for partial *pol* sequences (explanatory plot see Supplementary Fig. 1), we extracted 4, 11, 20, and 30 transmission clusters (8, 23, 44, and 74 patients) from the phylogeny of viral near full-length genome sequences in population A, and 12, 30, 40, and 61 transmission clusters (24, 65, 89, and 143 patients) from the phylogeny of partial *pol* sequences in population B.

**Heritability estimates for HIV-1 reservoir size**. We found a moderate but significant heritability of the HIV-1 reservoir size ~1.5 years after the initiation of ART using the viral near full-length genome sequences from population A (Fig. 2). Mixed-effect models yielded unadjusted heritability estimates which

**Table 1 Patient characteristics.**

|  |  | Population A | Population B |
|---|---|---|---|
| Sequenced HIV-1 genomic region |  | Near full-length | Partial *pol* |
| *n* |  | 351 | 610 |
| Age at first HIV-1 DNA sample, in years (median [IQR]) |  | 42 [37,48] | 43 [37,48] |
| Ethnicity (%) | White | 333 (94.87) | 566 (92.79) |
|  | Non-white | 18 (5.13) | 44 (7.21) |
| Sex (%) | Male | 307 (87.46) | 532 (87.21) |
|  | Female | 44 (12.54) | 78 (12.79) |
| Transmission group by sex (%) | MSM | 247 (70.37) | 403 (66.07) |
|  | HET male | 33 (9.4) | 76 (12.46) |
|  | HET female | 31 (8.83) | 50 (8.2) |
|  | PWID male | 22 (6.27) | 37 (6.07) |
|  | PWID female | 8 (2.28) | 20 (3.28) |
|  | Other male | 6 (1.71) | 17 (2.79) |
|  | Other female | 4 (1.14) | 7 (1.15) |
| Time of untreated HIV-1 infection, in years (%) | <1 | 43 (12.25) | 100 (16.39) |
|  | 1–3 | 51 (14.53) | 82 (13.44) |
|  | 3–5 | 90 (25.64) | 150 (24.59) |
|  | 5–7 | 61 (17.38) | 104 (17.05) |
|  | >7 | 106 (30.2) | 174 (28.52) |
| Time on ART at first HIV-1 DNA sample, in years (median [IQR]) |  | 1.5 [1.3,1.7] | 1.5 [1.3-1.7] |
| Time from ART initiation to below <50 HIV-1 RNA copies/ml, in years (median [IQR]) |  | 0.3 [0.2,0.5] | 0.3 [0.2,0.5] |
| CD4+ cell count pre-ART/μl blood (median [IQR]) |  | 215 [130, 286] | 214 [123, 299] |
| log 10 HIV-1 plasma RNA pre-ART/ml plasma (median [IQR]) |  | 5.0 [4.5, 5.4] | 4.9 [4.4,5.4] |
| HIV-1 RNA (180 days after ART initiation - 1st HIV-1 DNA sample) (%) | <50 copies/ml | 273 (77.78) | 472 (77.38) |
|  | Viral blips | 48 (13.68) | 82 (13.44) |
|  | Low level viremia | 27 (7.69) | 53 (8.69) |
| HIV-1 RNA (1st - 3rd HIV-1 DNA sample) (%) | <50 copies/ml | 232 (66.1) | 407 (66.72) |
|  | Viral blips | 88 (25.07) | 154 (25.25) |
|  | Low level viremia | 31 (8.83) | 49 (8.03) |

Patient characteristics in population A (with available viral near full-length NGS sequences obtained from HIV-1 plasma RNA) and population B (with available partial *pol* Sanger sequences obtained from HIV-1 plasma RNA).

The time of untreated HIV-1 infection was calculated using the estimated HIV-1 infection dates. Pre-ART log10 HIV-1 RNA copies/ml plasma and pre-ART CD4+ cell count/μl blood refer to the last laboratory values available before initiation of ART. Transmission group stratified by sex indicates the self-reported route of infection (men who have sex with men (MSM), heterosexual (HET), people who inject drugs (PWID), and other (including unknown, transfusions, and perinatal transmission)).

varied depending on the phylogenetic distance threshold, but which were consistently larger than zero (23–32%). Using the R package POUMM, we estimated the unadjusted heritability to be 24% [17%, 29%] based on the OU model and 10% [6%, 15%] based on the BM model. The discrepancy of these estimates is consistent with the known tendency of BM models (which are a special case of OU model ignoring stabilizing selection) to provide lower heritability estimates than the OU models[28]. Moreover, even the estimates using OU model should be interpreted as the lower bound for heritability, because any error due to poor fit of the OU model, noise in HIV-1 reservoir measurement and noise in the phylogeny will inflate the parameter $\sigma_e$ (scaled environmental variance), and hence implies that the obtained estimates underestimate the true heritability[28].

As clustered individuals tend to be clinically or demographically similar, unadjusted heritability estimates could potentially be inflated[41]. Accordingly, adjustment for potential clinical and demographical covariables lowered the heritability estimates for BM and OU models; however, it increased the estimates for the mixed-effect model: Upon adjustment for covariables, heritability estimates derived using near full-length genome sequences from the OU model decreased from 24 to 21%, stayed at 10% for the BM model and increased for all mixed-effect model thresholds to values between 29 and 78%. For partial *pol* sequences estimates decreased from 12 to 7% for the OU model, from 3 to 2% for the BM model, and from 69 to 57% for the strictest ME threshold. For all other ME thresholds, adjusting for covariables increased

the heritability estimates to values between 25 and 61% (Fig. 2). Generally, while broader confidence intervals for heritability were obtained in the adjusted models, adjusted and unadjusted estimates were qualitatively consistent for HIV-1 reservoir size. We further compared the adjusted heritability estimates with genetic information in different genes using the mixed-effect model. In this sensitivity analysis, large variations in heritability estimates using different genes and phylogenetic distance thresholds were observed (Supplementary Fig. 2); however, in general, the heritability estimates increased if the distance threshold was lowered.

Next, we estimated the heritability of the HIV-1 reservoir size in the larger study population (population B), in which partial *pol* sequences from Sanger sequencing were used (Fig. 2). Mixed-effect models yielded heritability estimates that were strongly dependent on the phylogenetic distance threshold, ranging from 24 to 69% (unadjusted) and from 25 to 57% (adjusted). The OU model yielded a lower heritability estimate of 12% [6%, 25%] (unadjusted) and 7% [3%,12%] (adjusted) and the BM model also yielded a lower estimate of 3% [2%,10%] (unadjusted) and 2% [1%,7%] (adjusted). Broadening our analysis to the full datasets with all HIV-1 subtypes included (Population $A_0$ and $B_0$) yielded overall slightly lower heritability estimates (Supplementary Fig. 3), which was expected considering the reduced homogeneity within the larger group. Further, heritability estimates from the phylogenies inferred from the overlapping fraction of population A and B yielded similar estimates, thus the difference in study

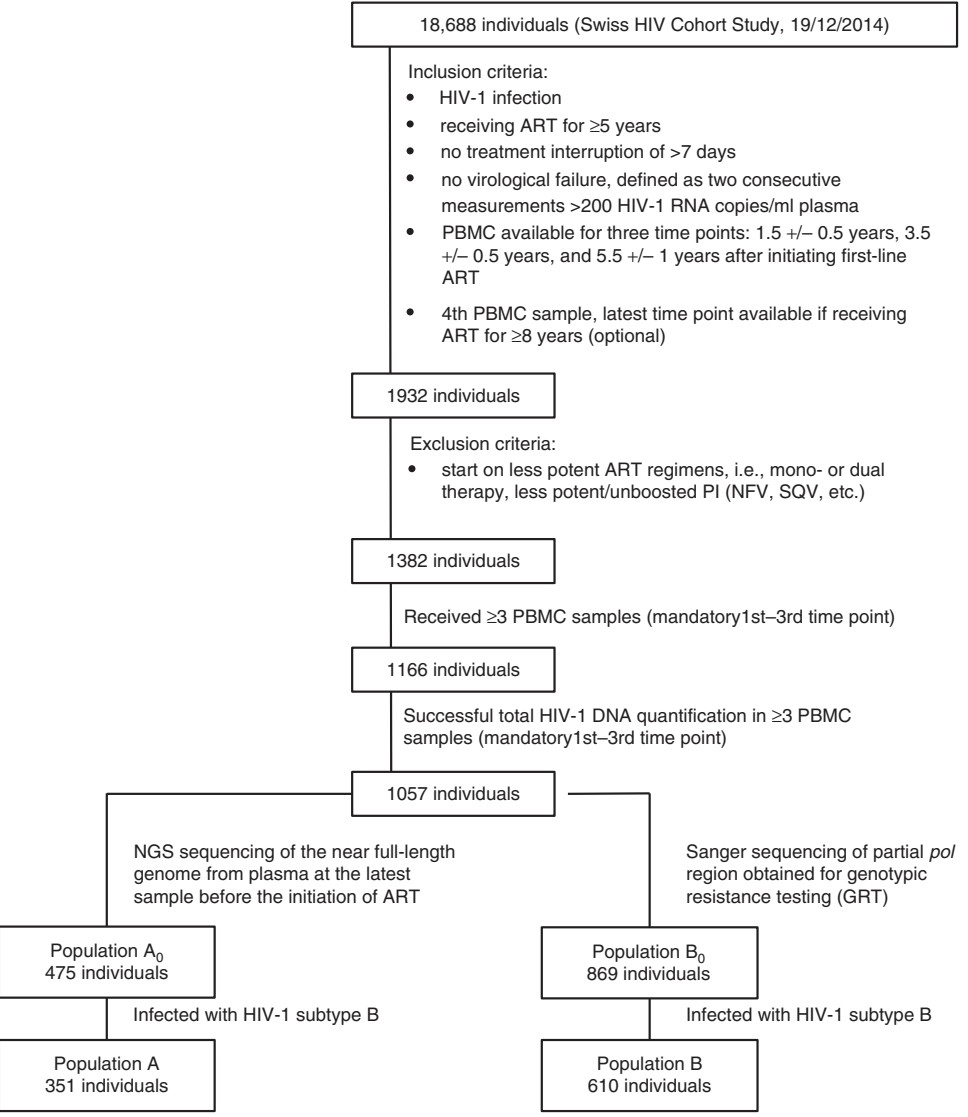

**Fig. 1 Patient inclusion flowchart.** ART antiretroviral therapy, PBMC peripheral blood mononuclear cells.

populations had no significant influence in our estimates (Supplementary Fig. 4). Generally, heritability estimates using partial *pol* sequences were qualitatively consistent with estimates using viral near full-length genome sequences.

Overall, we found that viral genetics explained a part of the HIV-1 reservoir size variability, which was dependent on the heritability estimator and dataset choice, but remained at 10% or above for all near-full HIV-1 genome-sequence approaches (Fig. 2). This indicated that the HIV-1 reservoir size was heritable in our study population.

**Heritability estimates for HIV-1 reservoir decay slope**. Using viral near full-length genome NGS sequences, we found tentative evidence that the HIV-1 reservoir decay slope was heritable in our study population A (Fig. 3) with estimates ranging from 3 to 85% (Fig. 3). The adjusted mixed-effect model estimates even ranged from 30 to 77%. The OU model yielded heritability estimates of 23% [9%, 40%] (unadjusted) and 10% [5%, 26%] (adjusted) and the BM model yielded heritability estimates of 3% [0%, 3%] (unadjusted) and 3% [1%, 4%] (adjusted). Similar to heritability estimates for HIV-1 reservoir size, adjustment for covariables increased the confidence intervals while yielding estimates comparable to the unadjusted values.

By contrast, using phylogenies inferred from partial *pol* Sanger sequences of population B, all unadjusted and adjusted heritability estimates were close to zero across different models (Fig. 3). Broadening our analysis to the full datasets with all HIV-1 subtypes included (Population $B_0$) yielded no relevant changes in heritability estimates derived using partial *pol* Sanger sequences (Supplementary Fig. 5).

The difference in heritability estimates using viral near full-length genome NGS sequences and partial *pol* Sanger sequences was not due to the difference in study populations (Supplementary Fig. 6). Further, using mixed-effect models, we compared the heritability estimates using phylogenies built from different genomic regions using near-full-length NGS sequences (Supplementary Fig. 7). The estimates using partial *pol* NGS sequences were zero for all thresholds, which was consistent with the estimates derived using partial *pol* Sanger sequences. However, we found positive heritability estimates using *gag* and *env* sequences. These estimates were above 13% except for the estimate using *gag* sequences under the most liberal mixed-effect model threshold. The difference across the genomic regions explained the overall higher estimates using near-full-length viral genome sequences compared to partial *pol* sequences.

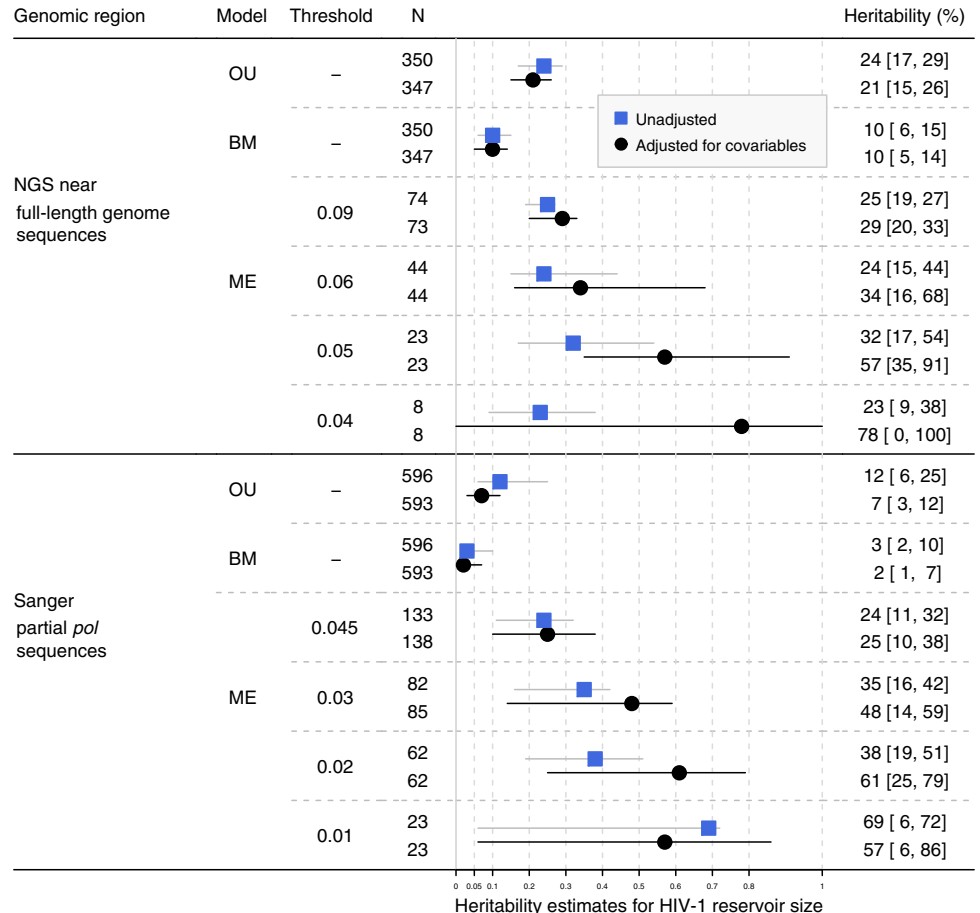

**Fig. 2 Heritability estimates for HIV-1 reservoir size based on the phylogenies built from near full-length HIV-1 genome NGS sequences and partial _pol_ Sanger sequences.** OU: Ornstein Uhlenbeck model. BM: Brownian motion model. ME: Mixed-effect model with corresponding phylogenetic distance threshold (substitutions per site). N: Number of patients included in the analysis. Patients with incomplete information of potential covariables were excluded. For BM and OU, all eligible patients from the tree were included while for mixed-effect model, only patients in the extracted transmission clusters were included. Black dots and black confidence intervals show the heritability estimates adjusted for covariables while blue rectangles and gray confidence intervals show the unadjusted estimates. 95% confidence intervals are shown in square brackets.

**Goodness of fit**. We compared the goodness of fit of all three models with the null model, which is given by a simple linear regression model assuming no correlation between patients on the phylogeny (Supplementary Tables 3 and 4). The null model would outperform other models if no heritability was present. In line with the reported heritability estimates, we found that all heritability estimators (i.e., BM, OU, or mixed-effect model) provided a better fit compared to the null model, for HIV-1 reservoir size using both sequence types (partial _pol_ and near-full length), and for HIV-1 reservoir decay using near full-length genome sequences. In these cases, the mixed-effect model with either the strictest or the most liberal threshold provided the lowest AIC, except for the unadjusted heritability estimate for HIV-1 reservoir size using partial _pol_ sequences, where the OU model provided the best fit. By contrast, for HIV-1 reservoir decay slope with partial _pol_ sequences, the null model provided the best fit, which indicated zero heritability in this case.

The comparison with the null model also provides an alternative approach to determine the significance of the viral heritability in the mixed-effect model (Supplementary Tables 5, 6): Depending on the phylogenetic distance threshold, different results were observed, which was likely due to the trade-off between the statistical power and the accuracy of the transmission cluster definition: Overall, in accordance with the positive heritability found for HIV-1 reservoir size using nearfull-length

genomes and partial _pol_ sequences and for reservoir decay using nearfull-length genome sequences, we observed significant random effects. By contrast, for reservoir decay and Sanger partial _pol_ sequences, where we reported no heritability, no significant random effects were found.

Among the phylogenetic mixed models, the difference between the fit of the BM and OU model was relatively small ($-3.01 < \text{AIC}_{\text{OU}} - \text{AIC}_{\text{BM}} < 3.44$) in our study. While the OU model provided a lower AIC compared to null model in some cases, we found that the likelihood surface was relatively flat across the model parameters, selection strength $\alpha$ and unit-time variance $\sigma$, in most cases (Supplementary Figs. 8, 9). The two parameters $\alpha$ and $\sigma$ were strongly correlated with each other. According to the Markov-Chain-Monte-Carlo (MCMC) uni-variate model density plots (Supplementary Figs. 8, 9), there was a close match between prior and posterior distributions for the parameters $\alpha$ and $\sigma$, while the posterior samples were notably distinct from the prior for the parameter $\theta$ (global optimum level) and $\sigma_e$ (scaled environmental variance). This suggested a lack of signal in some of our datasets for some of the OU parameters, i.e., OU models were in these cases not informed well enough by our data. Thus, we suggest being cautious about the parameter-estimates that are based on the OU models, in particular, for the selection strength parameter $\alpha$ and the unit-time variance parameter $\sigma$. Detailed discussion

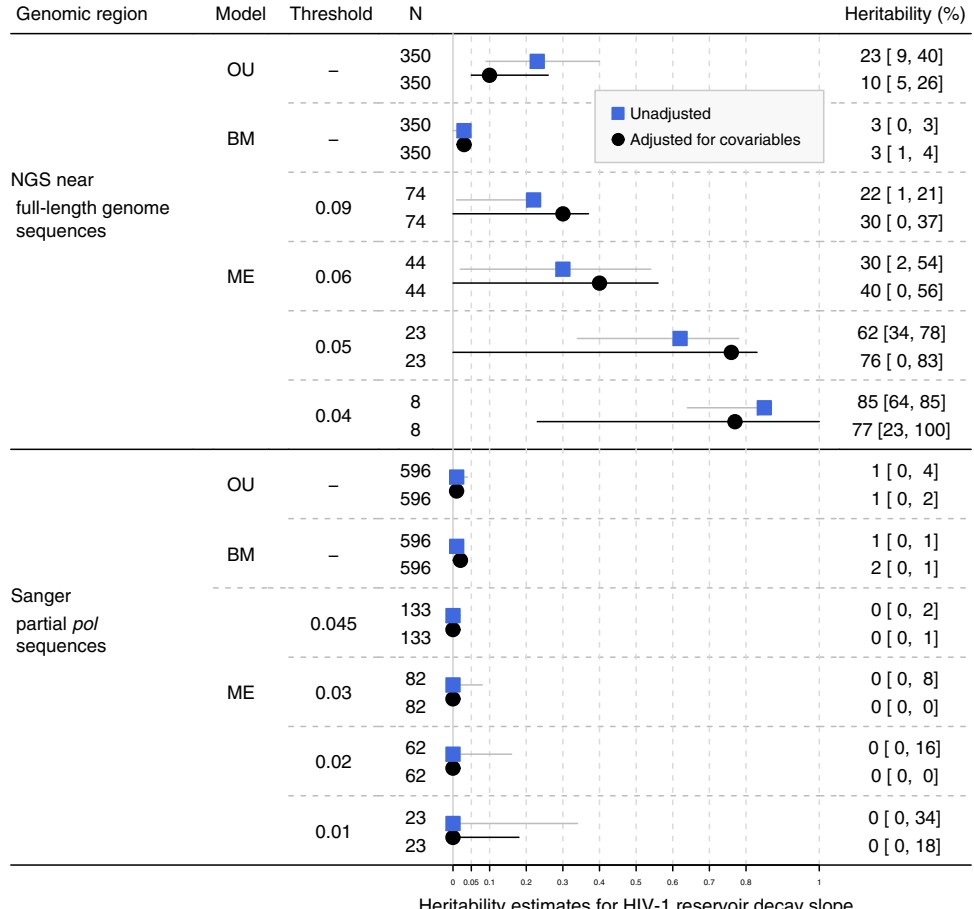

**Fig. 3 Heritability estimates for HIV-1 reservoir decay slope based on phylogenies built from near full-length HIV-1 genome NGS sequences and partial *pol* Sanger sequences.** OU: Ornstein Uhlenbeck model. BM: Brownian motion model. ME: Mixed-effect model with corresponding phylogenetic distance threshold (substitutions per site). N: Number of patients included in the analysis. Patients with incomplete information of potential covariables were excluded. For BM and OU, all eligible patients from the tree were included while for mixed-effect model, only patients in the extracted transmission clusters were included. Black dots and black confidence intervals show the heritability estimates adjusted for covariables while blue rectangles and gray confidence intervals show the unadjusted estimates. 95% confidence intervals are shown in square brackets.

and interpretation of the estimates from phylogenetic mixed models can be found in the Supplementary Discussion.

## Discussion

This work represents, to the best of our knowledge, the first study to estimate the heritability of the HIV-1 reservoir size and decay over extensive follow-up periods in a well-characterized population-based cohort of over 600 individuals infected with HIV-1 subtype B (Figs. 2, 3), as well as a broader population of over 800 individuals infected with diverse HIV-1 subtypes (Supplementary Figs. 3, 5). The size of our cohort enabled us to achieve estimates with high statistical support, while the extensive data of the underlying SHCS allowed to correct for many known and suspected confounders. We found that a moderate but significant fraction of the HIV-1 reservoir size at ~1.5 years after the initiation of ART was explained by viral genetic factors. Additionally, a tentative signal was also found for the decay of the viral reservoir 1.5–5.4 years after the initiation of ART using full-genome sequences, indicating that it was heritable in the corresponding study population. One major strength of our study is that the wide range of applied methods reached qualitatively consistent heritability estimates with overlapping confidence intervals between methods. In that sense, our findings were qualitatively robust with respect to potential confounding factors,

difference in sub-datasets analyzed, a wide range of phylogenetic distance thresholds, discrepancies between model assumptions, and potential sequencing errors.

Explicitly specifying and modeling environmental effects can reduce the upward bias in heritability estimates[42]. All methods used in our study allowed adjustment for the effects brought by viral and host characteristics. Generally, adjusting for covariables lowered or left unchanged heritability estimates derived with the OU and BM model in most cases (except the slope heritability estimate derived with the BM model using *pol* sequences) and increased heritability estimates derived with the ME model (except for heritability estimates derived with the strictest ME threshold). This indicates that overall, the inclusion of the covariables led to more complex models resulting in higher uncertainties, but reduced potential confounding by other covariables such as pre-treatment virus load which is known to affect the HIV-1 reservoir but can also cluster on the viral phylogeny.

Among potential viral and host characteristics that could influence HIV-1 reservoir size and its long-term dynamics, SPVL is expected to be of most importance. Consensus has been achieved that SPVL had a heritability of around 20–30%[28]. In our analysis, we used the last log 10 HIV-1 plasma RNA pre-ART/ml plasma values available before initiation of ART (RNApreART) as a substitute for SPVL for the following reasons: (1) SPVL can only be calculated if patients start treatment during chronic phase. In

our study, around 20% of patients started treatment during acute phase, so if SPVL was used, we would lose those 20% of patients. (2) More measurement and calculation errors can be introduced during the approximation and calculation of SPVL. (3) RNA-preART could be random across population considering the different stages where patients started treatment. However, as the heritability estimate for RNApreART in our dataset was 20.9%, and thus comparable to that of SPVL, we believe RNApreART can be used as an approximation for SPVL as a potential covariable[36].

As the heritability estimates from the mixed-effect model are highly dependent on the phylogenetic distance threshold, the trade-off between the accuracy of the transmission cluster definition and statistical power is to be considered when interpreting the results. While the liberal threshold could provide more statistical power because of a larger number of cluster members, the underlying assumption that little or no evolution occurred within clusters is more problematic with liberal thresholds and similarly the evidence that they are linked through transmission is weaker. Thus, considering a range of genetic distance thresholds is central to this study. Many HIV cluster studies have identified the proper thresholds for partial *pol* sequences[43,44], e.g., 0.045 substitutions per site as the upper bound, while no consensus has been achieved regarding near full-length genome sequences. Thus, we extrapolated the thresholds for near full-length genome sequences from partial *pol* sequences (Supplementary Fig. 1), assuming that both phylogenies shared the same partition of cherries by genetic distance. Overall, the wide range of distance threshold that we applied, yielded qualitatively consistent conclusions in our study, thereby underlying the robustness of our results.

Phylogenetic mixed models involve the evolutionary process of the trait according to a BM or OU process, respectively. More statistical power can be provided as all patients on the phylogeny are included in the analysis, compared with mixed-effect model which only includes patients from identified transmission clusters. However, we found a lack of signal in our dataset for some of the model parameters, especially, the selection strength parameter α and the unit-time variance parameter σ. Possible reasons include the following: First, traits related with HIV-1 reservoir may not undergo the assumed evolutionary process especially under suppressive ART, which can naturally explain why the model did not fit our data well. Second, as phylogenetic mixed model optimization requires large sample size for better and stable performance, especially for the OU model[36], our data size may have been too small compared to the ones that were used in previous studies for estimating heritability of SPVL (more than 3000 individuals). Finally, signals may be obscured by noise in the data introduced, for example, from incomplete and inaccurate genetic information (sequencing and polymerase chain reaction (PCR) errors) or from assay variability (total HIV-1 DNA measurements).

In our study, heritability estimates varied based on whether viral near full-length genome sequences or Sanger sequences were used. For both mixed-effect and phylogenetic mixed models, the difference which may influence heritability estimates lies in the reconstruction of a phylogenetic tree close to the real transmission networks using different genes. In principle, recombination can affect phylogenetic inference. Recombination may imply that different genes have different evolutionary history, thus phylogenies inferred from the near full-length genome sequences could be different from those inferred from partial *pol* genes[45]. A sensitivity analysis showed that effects from inter-subtype recombination did not disrupt our heritability estimates (see Supplementary Fig. 11), but still intra-subtype recombination could influence our estimates. Apart from recombination, phylogenetic inference could also be influenced by potential

convergent evolution[46], which describes the process that different individuals can acquire similar phenotypes or genotypes under similar environments, which can result in clustering of unrelated patients on the phylogeny. We excluded the common resistance mutation genes on *pol*, but mutations occurring on *gag* and *env* could still play a role, e.g., cytotoxic T lymphocytes (CTLs) escape mutations[47]. The extent to which these factors can influence the heritability estimates using different genes would merit further investigations.

In summary, the two datasets analyzed (near full-length genome sequences and partial *pol* sequences) represent a different compromise between evolutionary information and sampling density. Since heritability is a property of the population under consideration, no numerical accordance between estimates from the two populations should be expected. The advantage of the partial *pol* (Sanger) dataset is a higher sample size and higher sensitivity to obtain sequences also at lower viral loads due to the shorter amplified and sequenced fragment. The advantage of our NGS dataset is that it captures the near full-length HIV-1 genome, thereby allowing for a more exact phylogenetic reconstruction, which is methodologically more correct. However, it is a key strength of this study that different datasets were available, systematically assessed and compared. This may build the basis for further studies assessing the heritability of HIV-1 reservoir phenotypes without having access to different sequencing datasets.

Our study has some limitations. First, we used total HIV-1 DNA measured in peripheral blood mononuclear cells (PBMC) samples as a proxy for the latent HIV-1 reservoir. As total HIV-1 DNA levels are higher than the HIV-1 reservoir of replication competent latently infected cells, this is theoretically a limitation of our study. However, we believe our choice is the optimal compromise at this time: On the one hand, it has been shown that total HIV-1 DNA measured in PBMC samples is a sensitive, clinically relevant proxy for the HIV-1 reservoir[3,39,40,48] that also correlates well with the intact proviral DNA assay[40]. On the other hand, the very recently described, sophisticated methods (quantitative viral outgrowth assay, tat/rev induced limiting dilution assay etc) are not applicable to a cohort of over 600 longitudinally sampled participants, as they are too time- and labor intensive and/or require a high amount of cells, and thus, are only (prospectively) applicable to selected individuals[49,50]. Such studies of a few individuals, although highly interesting and important, always bear the risk of strong selection bias implying that generalizability to the population level is often problematic. In contrast, total HIV-1 DNA can be determined in large populations needed for heritability studies[3,39,48].Thus, there is a trade-off between the potential bias directly induced by the measurement method and the bias indirectly induced by method's restrictions and limitations of the study population.

In addition, even though the study population represents the largest cohort of patients with longitudinal HIV-1 reservoir measurements so far, statistical power remained a major limitation for several analyses: for the phylogenetic mixed model, the total number of patients in transmission clusters was relatively small compared with previous studies which estimated the heritability of other HIV-1 related traits[28]. This was in particular the case for the stricter distance threshold. As a consequence, we observed a large variability in heritability estimates across methods and phylogenetic distance thresholds. As our study is the first to determine the heritability of HIV-1 reservoir size and decay, the true ("ground truth") heritability of this trait remains unknown and thus, a final conclusion on model performance is impossible. Another limitation of our study is that the BM and OU model did not fit our data well, which kept us from drawing conclusions on the evolution of HIV-1 reservoir size and long-

term decay on population levels. Further, we did not analyze the impact of the diversity of viral populations on the HIV-1 reservoir phenotypes, which will merit further investigations in future studies. Finally, as for all molecular epidemiology and comparative-method studies, the validity of our conclusions depends on the accuracy of the phylogenetic inference from sequences, which could be influenced by many factors, such as within-host evolution, limited genetic information contained in short sequences (partial *pol*), recombination and convergent evolution. However, we do not expect this to have a substantial influence on our estimates and in particular our heritability estimates are conservative in the sense that phylogenetic error leads to an underestimation of heritability.

A great inter-individual variability in the number of latently HIV-1 infected cells and long-term dynamics under suppressive treatment was reported across different studies[5,7,51,52]. Quantifying the contribution of viral genetic factors to this variability is of importance both, from a basic and clinical science perspective, and may inform future reservoir and cure studies. In our study, we find evidence for a moderate but significant heritability of the HIV-1 reservoir size 1.5 years after initiation of ART and more tentative evidence for the heritability of the long-term dynamics of the HIV-1 reservoir decay. While the mechanisms underlying this heritability remain to be defined, our results indicate that the infecting HIV strain should be taken into consideration in future efforts to cure HIV.

## Methods

**The Swiss HIV Cohort Study**. The SHCS is a nationwide, prospective observational study founded in 1988 and enrolling HIV-infected adults of all transmission groups[53]. Clinical and laboratory data are collected every 3–6 months and plasma and cell samples are stored every 6–12 months. More than 75% of all HIV-1 infected individuals registered in Switzerland and receiving ART are enrolled in the SHCS[53]. The current study participants were included when they fulfilled the following inclusion criteria: (1) start on potent ART regimen (i.e., no mono- or dual therapy, no less potent/unboosted PI (NFV, SQV etc.), (2) no treatment interruption of >7 days, (3) no virologic failure as defined by two consecutive viral load measurements >200 HIV-1 RNA copies/ ml plasma, (4) available cell samples during ART, (5) either A. NGS sequencing of the full genome from plasma at the latest sample before the initiation of ART or B. Sanger sequencing of partial *pol* region obtained from plasma for genotypic resistance testing (GRT) and 6) infected with HIV-1 subtype B (Table 1, Fig. 1). The subtypes were determined using partial *pol* Sanger sequences. We additionally identified 12 potential recombinants using Comet[54] on near full-length genome sequences and did a sensitivity analysis excluding those patients (Supplementary Fig. 11). The SHCS has been approved by the ethics committee of the participating institutions and written informed consent was obtained from all participants.

**HIV-1 reservoir size and decay**. We focused on the following two distinct phenotypes which were pre-defined in the previous study[3]: 1. The HIV-1 reservoir size, i.e., the total HIV-1 DNA level 1.5 years after initiation of ART and 2. The HIV-1 reservoir long-term dynamics under ART, i.e., the decay slope of the total HIV-1 DNA from 1.5 to 5.4 years. Total HIV-1 DNA data were obtained from the dataset established in our previous study[3]. In this study PBMCs at ~1.5, ~3.5, and ~5.4 years after initiation of ART were collected, cellular DNA was extracted, total HIV-1 DNA was quantified, and reservoir decay rates were calculated (see subsections "Cells", "Genomic DNA extraction and fragmentation", and "Quantification of total HIV-1 DNA by droplet digital PCR" in the Methods section of ref. [3]). The distributions of HIV-1 reservoir size and decay in Population A and B are displayed in Supplementary Fig. 12.

**Phylogenetic tree construction**. We performed our heritability estimation for two distinct populations infected with HIV-1 subtype B (population A. of patients with viral near full-length genome NGS sequences available and population B. of patients with available GRT sequences) and accordingly with different phylogenies. For the construction of maximum likelihood phylogenetic trees, we included the partial *pol* Sanger sequences from the positions 2253–3870 and viral near full-length genome NGS sequences.

For partial *pol* sequences, we chose the sequence from the SHCS resistance database closest to the initiation of ART, if more than one sequence per patient was available, while near full-length genome NGS sequencing was performed systematically using the last plasma sample available before the initiation of ART for each patient. Sanger sequences were aligned to an HXB2 reference genome using MUSCLE[55]. We used consensus sequences, which were derived from NGS data

(determined as described in Supplementary Method) by the analysis pipeline V-pipe[56]. In this pipeline, NGS reads were preprocessed with PRINSEQ v0.20.4[57]. We then aligned the preprocessed reads to an HXB2 reference genome and generated the consensus sequences with a majority vote rule using ngshmmalign. Low coverage positions (<40 reads) were excluded and only NGS consensus sequences larger than 5000 bp were selected. For all sequences, insertions relative to HXB2 and resistance mutations according to the Stanford (http://hivdb.stanford.edu/) and International Antiviral Society-USA (https://www.iasusa.org/) lists were removed, and positions with many gaps were eliminated by trimAl[58]. Conserved blocks from multiple alignments were selected for phylogenetic analysis.

Phylogenetic tree construction was performed using the maximum likelihood algorithm RAxML version 8 with the GTRCAT model[59]. One hundred Bootstrap trees were constructed simultaneously with built-in options. Phylogenetic trees were inferred separately on five sets of sequences: the *gag*, *pol*, *env* and near full-length genome NGS sequences of population A and the partial *pol*-sequences of population B. All trees were rooted with an outgroup of HIV-1 subtype D sequences. To avoid the risk brought by rooting with distant outgroup, we additionally performed a sensitivity analysis using *LSD-0.2*[60] to find the root of the tree (Supplementary Figs. 13, 14).

**Extraction of transmission clusters**. We identified potential transmission clusters such that the maximum phylogenetic distance within the cluster was no larger than pre-defined threshold, using the R package APE[61] and custom scripts. For phylogenies built with partial *pol*-sequences, phylogenetic distance thresholds of 0.01, 0.02, 0.03, 0.045 substitutions per site were applied. For HIV-1 viral near full-length genome sequences, no consensus was achieved regarding the phylogenetic distance threshold for defining a transmission group. In our analysis, they were chosen such that they include the same percentage of cherries compared with using phylogeny built from partial *pol* Sanger sequences and applying cutoff of 0.01, 0.02, 0.03, and 0.045 (explanatory plot in Supplementary Fig. 2). The derived phylogenetic distance threshold for full-genome sequences were 0.04, 0.05, 0.06, 0.09 substitutions per site. Using similar methods, the derived thresholds for *gag* sequences were 0.01, 0.02, 0.03, and 0.05 and for *env* sequences these were 0.03, 0.05, 0.07, and 0.09 substitutions per site.

**Heritability estimates**. Heritability of the considered phenotypes was estimated with three models, mixed-effect model and phylogenetic mixed models with underlying Brownian motion or Ornstein Uhlenbeck process. Trait value data was fitted with a mixed-effect model using the R package NLME[62] (*lme* function, restricted maximum likelihood) and heritability was determined as the ratio of the between-cluster variance and the total variance.

Further, we applied phylogenetic mixed models to estimate the heritability of the HIV-1 reservoir traits. These models are widely used to estimate heritability from a phylogenetic tree, based on the assumptions that the trait evolved along the tree according to Brownian motion (BM) or Ornstein Uhlenbeck (OU) model respectively. In the BM model a trait is assumed to evolve according to the stochastic process $\sigma dW_t$, where $(W_t)_{(t \geq 0)}$ denotes BM and accounts for randomness in the divergence of a trait, and $\sigma$ scales the magnitude of fluctuations. The BM model is usually interpreted as a random unconstrained neutral evolution process. As an extension of Brownian motion, the Ornstein Uhlenbeck model adds an additional stabilizing selection towards the long-term optimal trait value. In the OU model, $(X_t)_{(t \geq 0)}$ is defined as

$$dX_t = \alpha(\theta - X)dt + \sigma dW_t,$$

where $\theta$ denotes the global optimum level and $\alpha$ the strength of selection.

The R package POUMM[28] was used for fitting the OU and BM model with Bayesian inference and maximum likelihood optimization, respectively. We reported in the main text the empirical (time-independent) heritability estimates $H_e^2$ from the package. As the package does not provide options to adjust for covariables, we performed a multivariate linear regression first to adjust for covariables and then took the residuals from the regression as trait value input into the optimization[37]. We also applied similar methods described by Blanquart et al.[36], which directly incorporated adjustment for covariables in maximum likelihood estimation for fitting the models (Supplementary Tables 3,4; Supplementary Figs. 8, 9). Using this method, the upper limit of $\alpha$ was set as 10, which was the same as implemented in Blanquart et al.[36].

Estimates from Bayesian inference were reported in the main text and estimates from maximum likelihood estimates were included in (Supplementary Tables 3, 4; Supplementary Fig. 8, 9). To assess the uncertainties in the phylogenetic tree interference and heritability estimation, all confidence intervals were derived based on the distribution of the estimates across the 100 bootstrapped trees.

**Adjustment for viral and host covariables**. Viral and host characteristics which were found to be associated with HIV-1 reservoir size and decay slope in previous work[3] or in the current population were adjusted for in the analysis. For HIV-1 reservoir size, we adjusted for transmission group, sex, ethnicity, age, time on ART, time to suppression, initiation of ART in acute/chronic infection, RNA and CD4 pre-ART, and prior viral blips. For reservoir decay slope, we adjusted for HIV-1 reservoir size, treatment center, time to suppression, CD4 pre-ART and viral blips.

We performed a sensitivity analysis to relax the linearity assumption made for continuous covariables. In these sensitivity analyses, we used polynomial splines (bs() function in the r package splines[63]) for all numerical variables (Supplementary Figs. 15, 16). Further, we performed a sensitivity analysis using an adjustment based on residuals of the full dataset ($N = 869$), thereby avoiding the inclusion of many covariables in an already small dataset and using the best information available in the full population for adjustment (Supplementary Figs. 17,18).

We defined viral load to be suppressed when all measurements were <50 HIV-1 RNA copies/ml plasma. We defined viral blips to be present when there were measurements ≥50 HIV-1 RNA copies/ml plasma, which were preceded and followed by measurements <50 HIV-1 RNA copies/ml plasma. Any subsequent viral load measurement ≥50 HIV-1 RNA copies/ml plasma within 30 days of a viral blip was considered to be part of the same viral blip[64]. Individuals who had multiple consecutive viral load measurements ≥50 HIV-1 RNA copies/ml plasma (without experiencing virological failure as defined by two consecutive viral load measurements >200 HIV-1 RNA copies/ml plasma) were considered to exhibit low-level viremia.

We determined the initiation of ART as occurring in acute/chronic infection using the estimated HIV-1 infection date, which was estimated using a hierarchical approach on the basis of indicators of varying reliability[65]. The following methods were used with decreasing priority to yield the maximal accuracy for HIV-1 infection dates possible:

1. Defined HIV-1 primary infection: Either seroconversion dates (negative and positive HIV-1 screening tests less than 1 year apart) or a diagnosis of a primary infection were available as previously described[66]. We used the midpoint between the dates of the negative and positive tests or, if known, the date of the primary infection as the estimated HIV-1 infection date for these individuals.

2. Defined recent HIV-1 infection: If genotypic HIV-1 drug resistance test within the first year after diagnosis revealed low HIV-1 diversity (less than 0.5% of ambiguous nucleotides) and CD4+ cell counts were >200 cells/μl blood at registration[43,67,68], we interpreted these as recent HIV-1 infections and used the diagnosis date as an estimate for the infection date.

3. HIV-1 infection date estimates based on a back-calculation method using CD4+ cell counts and their slopes when available[69].

4. For the remaining individuals, no accurate dating was available. For these individuals the date of diagnosis was used as substitute for the HIV-1 infection date, which allowed us to define at least the minimum length of HIV-1 infection.

Time to viral suppression was defined as the time from initiation of ART until the first viral load below 50 copies/ml HIV-1 plasma RNA.

HIV-1 subtype was determined using the REGA HIV-1 subtyping tool[70] and COMET[54].

**Reporting summary.** Further information on research design is available in the Nature Research Reporting Summary linked to this article.

## Data availability

The individual level datasets generated or analyzed during the current study do not fulfill the requirements for open data access: (1) The SHCS informed consent states that sharing data outside the SHCS network is only permitted for specific studies on HIV infection and its complications, and to researchers who have signed an agreement detailing the use of the data and biological samples; and (2) the data are too dense and comprehensive to preserve patient privacy in persons living with HIV.

According to the Swiss law, data cannot be shared if data subjects have not agreed or data are too sensitive to share. Investigators with a request for the data that support the findings of this study should contact the corresponding author Roger D. Kouyos and the Scientific Board of the SHCS. The provision of data will be considered by the Scientific Board of the SHCS and the study team and is subject to Swiss legal and ethical regulations, and is outlined in a material and data transfer agreement.

## Code availability

All code generated during the current study can be made available from the corresponding author on request.

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

## Acknowledgements

The authors thank the patients for participating in the SHCS, the study nurses and physicians for excellent patient care, A. Scherrer, A. Traytel, and S. Wild for excellent data management and D. Perraudin and M. Amstutz for administrative assistance. This work was funded within the framework of the Swiss HIV Cohort Study (SNF grant #33CS30_177499 to H.F.G). The data were gathered by the Five Swiss University Hospitals, two Cantonal Hospitals, 15 affiliated hospitals and 36 private physicians (listed in http://www.shcs.ch/180-health-care-providers). The work was furthermore supported by the Systems.X grant #51MRP0_158328(to N. Bachmann, J. Bogojeska, JF., V.R., R.K., H. F.G., K.J.M.), by SNF grant 324730B_179571(to HFG), SNF grant SNF 310030_141067/1 (to H.F.G. and K.J.M.), SNF grants no. PZ00P3-142411 and BSSGI0_155851 to R.D.K., the Yvonne-Jacob Foundation (to H.F.G.), the University of Zurich's Clinical Research Priority Program viral infectious disease, ZPHI (to H.F.G) and the Vontobel Foundation (to H.F.G. and K.J.M.).

## Author contributions

N. Beerenwinkel, J. Bogojeska, J.F., V.R., K.J.M., H.F.G., and R.D.K. conceived and designed the study and analyzed data. K.N. and K.J.M. designed and performed experiments and analyzed data. C.W., N. Bachmann, V.M., F.B., S.P.C., T.T., and R.D.K. conducted computational analyses and contributed analysis tools and data analysis. J. Böni, M.P., T.K., S.Y., M.B., L.W., A.C., P.V., E.B., M.C., H.F.G. and the members of the Swiss HIV Cohort Study conceived and managed the SHCS and ZPHI cohorts collected and contributed patient samples and clinical data. C.W., N. Bachmann and R.D.K. wrote the manuscript, which all coauthors commented on.

## Conflict of interest

M.C. has received research and travel grants for his institution from ViiV and Gilead. E. B. has received fees for his institution for participation to advisory board from MSD, Gilead Sciences, ViiV Healthcare, Abbvie and Janssen. M.B. has received research or educational grants by Abbvie AG, Gilead Sciences Switzerland Sàrl, Janssen-Cilag AG, MSD Merck Sharp & Dohme AG and ViiV Healthcare GmbH. T.K. has received honoraria from Gilead Sciences and Roche Diagnostics. L.W. has received honoraria for advisory boards and/or travel grants: MSD, Gilead Sciences. All remuneration went to her home institution and not to L.W. personally. R.D.K. has received grants from the Swiss National Science Foundation and personal fees from Gilead Sciences, outside the submitted work. H.F.G. has received unrestricted research grants from Gilead Sciences and Roche; fees for data and safety monitoring board membership from Merck; consulting/advisory board membership fees from Gilead Sciences, Merck, ViiV, Sandoz and Mepha. K.J.M. has received travel grants and honoraria from Gilead Sciences, Roche Diagnostics, GlaxoSmithKline, Merck Sharp & Dohme, Bristol-Myers Squibb, ViiV and Abbott; and the University of Zurich received research grants from Gilead Science, Roche, and Merck Sharp & Dohme for studies that Dr. Metzner serves as principal investigator, and advisory board honoraria from Gilead Sciences. All other authors have no potential conflict of interest.

## Additional information

## the Swiss HIV Cohort Study

Alexia Anagnostopoulos[1], Manuel Battegay[13], Enos Bernasconi[16], Jürg Böni[2], Dominique L. Braun[1], Heiner C. Bucher[21], Alexandra Calmy[12], Matthias Cavassini[17], Angela Ciuffi[22], Günter Dollenmaier[23], Matthias Egger[24], Luigia Elzi[12], Jan Fehr[1], Jacques Fellay[6,7], Hansjakob Furrer[13], Christoph A. Fux[25], Huldrych F. Günthard[1,2], David Haerry[26], Barbara Hasse[1], Hans H. Hirsch[10,12], Matthias Hoffmann[15], Irene Hösli[27], Michael Huber[2], Christian Kahlert[14,28], Laurent Kaiser[11], Olivia Keiser[24], Thomas Klimkait[11], Roger D. Kouyos[1,2], Helen Kovari[1], Bruno Ledergerber[1], Gladys Martinetti[29], Begona Martinez de Tejada[30], Catia Marzolini[12], Karin J. Metzner[1,2], Nicolas Müller[1], Dunja Nicca[14], Paolo Paioni[31], Guiseppe Pantaleo[9], Matthieu Perreau[10], Andri Rauch[14], Christoph Rudin[32], Alexandra U. Scherrer[1,2], Patrick Schmid[14], Roberto Speck[1], Marcel Stöckle[12], Philip Tarr[33], Alexandra Trkola[2], Pietro Vernazza[14], Gilles Wandeler[13], Rainer Weber[1] & Sabine Yerly[12]

[21]Basel Institute for Clinical Epidemiology and Biostatistics, University Hospital Basel, University of Basel, Basel, Switzerland. [22]Institute of Microbiology, University Hospital Lausanne, University of Lausanne, Lausanne, Switzerland. [23]Centre for Laboratory Medicine, Canton St. Gallen, St. Gallen, Switzerland. [24]Institute of Social and Preventive Medicine, University of Bern, Bern, Switzerland. [25]Clinic for Infectious Diseases and Hospital Hygiene, Kantonsspital Aarau, Aarau, Switzerland. [26]Positive Council, Zurich, Switzerland. [27]Clinic for Obstetrics, University Hospital Basel, University of Basel, Basel, Switzerland. [28]Children's Hospital of Eastern Switzerland, St. Gallen, Switzerland. [29]Cantonal Institute of Microbiology, Bellinzona, Switzerland. [30]Department of Obstetrics and Gynecology, University Hospital Geneva, University of Geneva, Geneva, Switzerland. [31]University Children's Hospital, University of Zurich, Zurich, Switzerland. [32]University Children's Hospital, University of Basel, Basel, Switzerland. [33]Kantonsspital Baselland, University of Basel, Basel, Switzerland.

