## [Peer Review File · Nature Communications]

Reviewers' Comments:

Reviewer #1:

Remarks to the Author:

The authors present an original and interesting work on the heritability of the HIV-1M reservoir size and decay rate. My main concerns relate to the way the authors have estimated the phylogenies and to analysing the data with a BM or OU model in a single HIV-1 group M phylogeny, which may not be the best way to go about it.

Major comments:

1) recombination breaks the bifurcating tree assumption. For this reason the authors should analyse non-recombinants and recombinants separately. For example, it seems that the authors have analysed subtypes A and G in a single phylogeny, which is not advised (see Abecasis et al 2007, Lemey et al 2009). Obviously, this also holds for other recombinant forms.

2) rooting with distant outgroup is a risky practice (see e.g. ebola, Dudas & Rambaut 2014, Gire et al 2014). In this case the long branch to the SIV outgroup can impact the clustering within HIV-1 group M. Why did the authors not choose for rooting based on the clock model assumption instead - there are good softwares available in the ML framework (treedater, LSD, treetime)?

3) I see several reasons why analysing the data in a single phylogeny is perhaps not ideal:

a) what if the different subtypes have different reservoir size/decay optima? This cannot be captured with current OU models (single optimum). Related to this: the estimates under the ME model are usually higher when compared to the estimates of the BM and OU model. Does this not suggest that there may indeed be different optima for the subtypes? How do the authors explain this tendency for the ME model?

b) related to the BM model: the groupM phylogeny is rather star-like akin to the null-model of no heritability. Same holds for subtype-specific data sets - these too will yield starlike phylogenies. What if the authors focus on the subset of the data that share a more recent common ancestry (better approximate transmission chains by subtype); perhaps then the signal for heritability becomes more clear? How would this affect the OU model? I anticipate that this can make the estimates more accurate.

c) analysing the groupM data in a single phylogeny brings about a higher risk of breaking the constant variance assumption underlying BM

4) The authors make a case for the data not always being very informative. Hence, stating that there is 'clear' evidence for a heritability component is over-interpreting the results (end of Discussion section).

Minor remarks:

1. introduction line 85: for the sake of completeness, please also cite Vrancken et al, MEE 2015.

2. the partial pol and NFL data sets represent a different compromise between evolutionary information (tree accuracy) and sampling density. I may have overlooked it, but I believe that it is good to make this explicit.

Reviewer #2:

Remarks to the Author:

The manuscript entitled "Heritability of the HIV-1 reservoir size and decay under long-term suppressive ART" by Wan et al. studies a very important question in HIV dynamics: what control the reservoir size and decay in patients undergoing suppressive ART; in particular, whether there is measurable heritability in this trait. Since the size and persistence of the reservoir are thought to control prognosis, this is an important topic. Unfortunately, this reviewer is left unconvinced about the conclusions presented from the data and the analysis presented in the manuscript.

The central issue is, of course, confounding. Since HIV is transmitted by contact (e.g., needle-sharing or sexual), and human contact is decidedly nonrandom, the recipient of the infection is often more similar to the donor than to a random person. Thus, naive measures of heritability may reflect properties of the transmission network, rather than of the underlying viral biology that this work claims to be characterizing (vide the last sentence of the abstract "Our findings indicate that viral genetic factors contribute to the HIV-1 reservoir size and hence the infecting HIV-1 strain may affect individual patients' hurdle towards a cure of HIV.")

The authors are, of course, aware of this and try to correct for it by mixed effect models applied to a breakup of the data into transmission clusters; and then controlling for covariates. Not surprisingly, they find that the estimation errors increase as the threshold for transmission cluster is made more stringent (though they do not discuss whether this is roughly the expected sort-sample-size increase), but, problematically, they also find that the covariance-adjusted point estimate of the heritability of the reservoir size in the NGS analysis tends to zero as the definition is made more stringent. Even more surprisingly, the trend is reversed if we look at the phylogeny from the more conserved Pol region. This calls for a comparison of the two sets of phylogenies (not only of their shapes in terms of distribution of 'cherries'), something not presented in the manuscript.

The phylogenetically corrected methods are more interesting once corrected for the covariates. Here one finds that the better determined method that assumes an undirected Brownian evolution finds no effect. A more complex model (that seems to be chosen by AIC and a not-fully-described likelihood ratio method, but not by BIC) seems to find a significant effect (the control for covariates in this method is indirect, though possibly adequate). The problem, however, is that the method may be ill-suited when the sample is mixed for subtype (since it forces a single "optimal" value on all sequences), and in the actual implementation, it encountered flat-likelihood directions, and in some cases the posterior distribution was equal to the prior. In a situation like this, it is not a priori clear whether the results are being forced by the choice of these other priors, or represents a true effect.

I have a few minor comments as to presentation. In many places the manuscript reads a bit imprecise: e.g., 'different methods yielded more comparable heritability estimates' (what does "more comparable" mean?) or "Generally, adjusting for potential covariables lowered our heritability estimates in most cases, which indicated that the part of trait variation which was originally attributed to genetic factors when no covariables were specified was actually explained by those chosen covariables." (what does "most cases" mean: this is not what I see in the figures). Also, I do not see that "Overall, we found that viral genetics explained a moderate but significant fraction of the HIV-1 reservoir size variability independent of model choice and robust to the adjustment for covariables." is an accurate description of the state of affairs. At other places, the statements are difficult to parse: "However, heritability estimates larger than zero were found using the genetic information contained in gag and env genes." (how exactly did the genetic information, rather than their variation informing phylogenetic inference, used to deduce heritability?)

My overall conclusion is that this is data that would be of value to the field, but the claims made in the manuscript seem overly strong.

Tanmoy Bhattacharya (tanmoy@lanl.gov)

Reviewer #3:

Remarks to the Author:

The understanding of the reservoir size is important to eradication or functional cure of HIV-1 infection. Wan et al studied over 800 treated patients and use the sequences generated by the Sanger or NGS method from plasma samples closest to the initiation of ART to predict the heritability of HIV-1 reservoir size and decay. Both pol and near full-length genomes sequences were studied using three different algorithms, and many factors that can affect the results were adjusted, although how they were adjusted was not clearly stated. They found a moderate but significant fraction of the reservoir size 1.5 years after ART was explained by genetic factors and less significant for the reservoir decay. While the results seems interesting, the study has serious issues with a number of major concerns. The lack of experimental details also make the manuscript very difficult to understand. It is not clear how the results will help to reduce the reservoir size for better clinical outcomes.

The total DNA level is much bigger than the real reservoir size in each patient. It is not the same as real reservoir and not a reliable proxy for the reservoir size.

The critical conclusions "moderate but significant" and "more tentative" were not well justified. No p values could be found for either conclusion.

It is unclear what phylogenetic information of HIV-1 sequences was used to predict heritability of the reservoir size. Are the lengths of the branches? It is likely so since only one sequence was generated by the Sanger method. Since the lengths of infection will have significant effects on branch lengths in phylogenetic trees, the uncertain infection time that were not well defined among most of the cases is a big concern. Will the ambiguous bases in the sequences from chronically infected patients affect the results, especially when a large number of sequences are analyzed together? If the branch lengths are an major factor, will different subtypes have more impact on reservoir sizes; ie different subtypes have huge different in reservoir sizes? It is unclear how the NGS data were used. Is the consensus sequence were generated for each patients? Again, how the quasispecies populations of chronic viruses were treated?

Whether the sequences were generated from just before ART or not should not matter if the heritability is determined by the nature of the HIV-1 sequences. If not, the timing will be difficult to control.

The huge variation among three different algorithms and among different thresholds for the ME method is troubling. What is the consensus? Which is more reliable to use?

It is unclear how the percentages of heritability were obtained. Is 18% too high or 2% too low? Which one is more accurate?

Different gene sequences and whole genome sequences have different predicative values. It is not clear which should be more reliable to use.

Why is the transmission cluster important for the analysis?

The actual slope data should be shown for readers to better understand the data for figure 3. 5.4 years do not seem long term.

Point by point reply

We are grateful for the reviewers' positive assessment of our work and appreciate their thoughtful comments and constructive suggestions. We have addressed all points raised by the reviewers and believe that this has substantially strengthened the manuscript. Our point-by-point reply contains reviewer's comments in blue, our answers in black, and changes in the manuscript in brown.

We would like to highlight first the following **main points of our revisions**:

1. Due to the very valid concerns of all three reviewers regarding the inclusion of different HIV-1 subtypes in the same analysis, we now restricted the analysis shown in the main text to HIV-1 subtype B. For completeness, we still show the original analysis including all HIV-1 subtypes in the Supplementary material (*Figure S3 and S5*). While this new analysis strengthened considerably the robustness of our heritability estimates, it lowered our sample size and thereby slightly increased the confidence intervals.
2. This decrease in total sample size lead to a small number of datapoints in the adjusted mixed-effect models with the strictest threshold. While it is worth noting that in the adjusted models several covariables were constant for the subpopulations considered, we still verified that the results of these analyses were still meaningful by added a new sensitivity analysis. We performed a linear regression analysis on our full dataset (N = 869) and used the residuals thereof for the adjustment for confounders in the mixed-effect models. This approach harbors the advantages of (i) avoiding the inclusion of many covariables in an already small dataset and (ii) using the best information available in the full population for adjustment. *Figure S17 and S18* display this sensitivity analysis, which indicates only minor differences to our original analysis.
3. We performed several sensitivity analyses regarding rooting of the tree, exclusion of suspected recombinants and dependence on sample size. Those analyses underline the robustness of our results and have been integrated into the main text or the Supplementary material.
4. When using mixed-effect models as heritability estimators, there is an intrinsic tradeoff between statistical power and methodological accuracy. While a higher sample size (and thereby higher statistical power) is achieved by more liberal distance thresholds, methodologically the heritability estimator is more accurate for strict thresholds. As this previously led to confusion, we discuss this point extensively in the revised version of the manuscript and added a sensitivity analysis (*Figure S19*) to the Supplementary material demonstrating the effect of decreasing sample size on the heritability estimates.
5. In the revised version of the manuscript, we further emphasized that it is a key strength of this study that it provides a systematic assessment of heritability estimates for HIV-1 reservoir size and decay dynamics by applying different heritability estimation methods to distinct datasets (i.e., HIV-1 sequences obtained using NGS and Sanger sequencing). While it is unavoidable that in such an analysis the numerical heritability value differs across estimation methods and datasets (in particular when sample size becomes small and hence sampling effects become strong - please see also point 4 above), we believe that this systematic assessment strengthens our conclusions by showing a consistent pattern of a substantial heritability of the HIV-1 reservoir size but weaker patterns for the HIV-1 decay slope.
6. Adding to point 5, we now highlight that among the heritability estimators each method has advantages and disadvantages—accordingly, no method is clearly superior compared to the other methods. The mixed effect (ME) models intrinsically are subject to a tradeoff between statistical power and methodological accuracy (see

point 4). However, compared to Ornstein-Uhlenbeck (OU) and Brownian Motion (BM) models they have the advantage of not assuming any evolutionary model of trait evolution. At the same time, OU and BM model have the advantage of utilizing the phenotype information of all tips on the phylogeny; compared to ME model, only considering the phenotype information of clustered tips. Between OU and BM models; OU models are theoretically clearly superior, accounting for a global phenotype optimum and stabilizing selection. However, these additional model parameters lead to a computational disadvantage of the OU models in the optimization process. While it may seem unsatisfactory that we cannot provide a single, definitive estimate of the HIV-1 reservoir size and decay heritability, our results represent the best available evidence that is currently achievable for the heritability of HIV-1 reservoir phenotypes, considering that our in depth analysis is based by far on the largest and most detailed dataset on this topic.

Reviewer #1 (Remarks to the Author):

The authors present an original and interesting work on the heritability of the HIV-1 M reservoir size and decay rate. My main concerns relate to the way the authors have estimated the phylogenies and to analysing the data with a BM or OU model in a single HIV-1 group M phylogeny, which may not be the best way to go about it.

Reply: We thank the reviewer for the positive assessment of our work and the valuable feedback. Regarding the concerns raised, please see our detailed responses and implemented changes outlined below.

1. recombination breaks the bifurcating tree assumption. For this reason the authors should analyse non-recombinants and recombinants separately. For example, it seems that the authors have analysed subtypes A and G in a single phylogeny, which is not advised (see Abecasis et al 2007, Lemey et al 2009). Obviously, this also holds for other recombinant forms.

Reply: We would like to thank the reviewer for this valuable comment, which is in line with comments of reviewer 2 (comment 3) and reviewer 3 (comment 5) on the inclusion of different HIV-1 subtypes in our phylogeny.

General Considerations: While we believe that studying a real-world diverse population is a major strength of our cohort in principle, we fully agree that we did not sufficiently consider the risk of fitting a global HIV-1 reservoir size/decay optimum for Ornstein-Uhlenbeck (OU) models to our phylogeny. Indeed, the HIV-1 reservoir might be partly dependent on the HIV-1 subtype: in our previous study, we observed trends of lower HIV-1 reservoir size (although statistically non-significant) and faster HIV-1 reservoir decays in individuals infected with HIV-1 non-B subtypes¹. Further, here, we did not take into account that building a mixed-subtype tree could have inflated internal branch lengths and thereby might have lowered OU and Brownian motion (BM) estimates.

These concerns, however, should not impact the mixed-effect (ME) model heritability analysis presented in the manuscript, as it focuses on transmission cluster similarities and no global optimum is fitted. ME model estimations have the advantage of depending less on the validity of model assumptions for the evolution of the trait considered and also less on (small) errors in the phylogenetic reconstruction (this was actually the main reason why we extensively tested the validity of ME-model-estimators in a recent publication²). However, since ME models are not clearly superior in other terms (please see our main point 6 of revisions) and to provide the reader with a full picture of heritability estimates of HIV-1 reservoir size and decay, we aimed to show results for a variety of heritability estimators³.

In summary, we completely agree with the reviewer and adapted the analysis accordingly:

Adaptation of data analysis: Since the sample size of HIV-1 non-B subtypes was too small to analyze sub-trees per HIV-1 subtype (no optimization was possible), we decided to focus our main analysis on HIV-1 subtype B. Please find below the results shown in the new figures, which will replace *Figure 2* and *Figure 3* in our main text. Compared to our first analysis including all HIV-1 subtypes of group M, heritability estimates increased from 14% to 24% for the OU model (NGS, adjusted for covariables), from 1% to 10% (now significant) for the BM model (NGS, adjusted for covariables) and from 22%, 22%, 34%, and 0% to 29%, 34%, 57%, and 78%, respectively, for the ME model (NGS, adjusted for covariables, with threshold 0.09, 0.06, 0.05, and 0.04, respectively). Note however, that the ME model with the strictest threshold was uninformative, due to the even smaller sample size, as the confidence interval spanned from 0 to 1. The increase in heritability estimates for the BM and OU models may reflect that the phylogeny does no longer contain overly long internal branches, which likely lowered our previous heritability estimates. Further, for the OU models this increase in heritability estimates may also reflect the effect of not fitting anymore a single optimum to different HIV-1 subtypes (as suggested by the reviewer). We wish to thank the reviewer again for this comment, as it has helped to substantially strengthen our results.

Finally, we also reanalyzed the HIV-1 subtypes using Comet⁴ on near full-length HIV-1 genome sequences and did a sensitivity analysis excluding the 12 sequences which appeared to be recombinants (*Figure S11*). Differences in heritability estimates when excluding these 12 sequences were small and in line with what would be expected from reducing the sample size randomly by 12 sequences (see *Figure S11*).

Adaptations in the manuscript: Due to the considerations mentioned above, we restricted the core dataset for analysis in the manuscript to HIV-1 subtype B and show the original analysis including all subtypes only in the supplementary material. This resulted in a change of all figures and all numbers presented throughout the entire manuscript. We therefore marked all the changes using the track-changes mode in the manuscript (instead of listing them in addition in the point-by-point reply and in order to keep the point-by-point reply as concise as possible). “We explained the restriction in the description of our study population, line 119:

“Considering the inter-subtype heterogeneity of HIV-1, we restricted our study populations to individuals infected with HIV-1 subtype B strains, which were 351 individuals with available NGS sequences (denoted as population A) and 610 individuals with available Sanger sequences (denoted as population B) (**Fig 1**).

[...]

We additionally performed analyses on the full datasets (population A₀ and B₀) including all HIV-1 subtypes (**Supplementary Table S1**).”

Figure 2 (HIV-1 subtype B only, replaced in manuscript). Heritability estimates for HIV-1 reservoir size with phylogeny built from near full-length HIV-1 genome NGS sequences and partial pol Sanger sequences. OU: Ornstein Uhlenbeck model. BM: Brownian motion model. ME: Mixed-effect model with corresponding phylogenetic distance threshold (substitutions per site). N: Number of patients included in the analysis. Patients with incomplete information of potential covariables were excluded. For BM and OU, all eligible patients from the tree were included while for mixed-effect model, only patients in the extracted transmission clusters were included. 95% confidence interval shown in square brackets.

Figure 3 (HIV-1 subtype B only, replaced in manuscript). Heritability estimates for HIV-1 reservoir decay slope with phylogeny built from near full-length HIV-1 genome NGS sequences and partial pol Sanger sequences. OU: Ornstein Uhlenbeck model. BM: Brownian motion model. ME: Mixed-effect model with corresponding phylogenetic distance threshold (substitutions per site). N: Number of patients included in the analysis. Patients with incomplete information of potential covariables were excluded. For BM and OU, all eligible patients from the tree were included while for mixed-effect model, only patients in the extracted transmission clusters were included. 95% confidence interval shown in square brackets.

Figure S11. Heritability estimates adjusted for covariables using the phylogeny built from near full-length HIV-1 genome NGS sequences: Sensitivity analysis excluding potential recombinants identified by Comet. Black points (“Baseline estimates”) and confidence intervals show the estimates presented in Figure 2 and 3 of the manuscript, violet squares and gray confidence intervals show the corresponding estimates after exclusion of recombinants identified by Comet.

2. rooting with distant outgroup is a risky practice (see e.g. ebola, Dudas & Rambaut 2014, Gire et al 2014). In this case the long branch to the SIV outgroup can impact the clustering within HIV-1 group M. Why did the authors not choose for rooting based on the clock model assumption instead - there are good softwares available in the ML framework (treedater, LSD, treeTime)?

Reply We would like to thank the reviewer for indicating this very relevant point. We performed another sensitivity analysis using *LSD-0.2*⁵ to find the root of the tree, a comparison is shown in our new *Figures S13 and S14*. Heritability estimates were very close for NGS and Sanger sequences for the phenotype HIV-1 DNA levels 1.5 years after initiation of ART as well as for HIV-1 DNA decay slope 1.5-5.4 years after initiation of ART. Thus, we decided not to change the main analysis, but to highlight the new supplementary analyses in line 438 of the Methods section:

“To avoid the risk brought by rooting with distant outgroup, we additionally performed a sensitivity analysis using *LSD-0.2* to find the root of the tree (**Supplementary Fig S13-14**).”

Figure S13. Heritability estimates adjusted for covariables using different rooting methods for the HIV-1 reservoir size 1.5 years after initiation of ART. Black dots (“Rooting with HIV-1 subtype D”) and black confidence intervals show the estimates presented in Figure 2 of the manuscript. Light blue rectangles (“Rooting with LSD”) and grey confidence intervals show the estimates from the phylogeny rooted with the root position found by LSD.

Figure S14. Heritability estimates adjusted for covariables using different rooting methods for the HIV-1 reservoir decay slope 1.5-5.4 years after initiation of ART. Black dots (“Rooting with subtype D”) and black confidence intervals show the estimates presented in Figure 3 of the manuscript. Light blue squares (“Rooting with LSD”) and grey confidence intervals show the estimates from the phylogeny rooted with the root position found by LSD.

3. I see several reasons why analysing the data in a single phylogeny is perhaps not ideal: a) what if the different subtypes have different reservoir size/decay optima? This cannot be captured with current OU models (single optimum). Related to this: the estimates under the ME model are usually higher when compared to the estimates of the BM and OU model. Does this not suggest that there may indeed be different optima for the subtypes? How do the authors explain this tendency for the ME model?

b) related to the BM model: the groupM phylogeny is rather star-like akin to the null-model of no heritability. Same holds for subtype-specific data sets - these too will yield star like phylogenies. What if the authors focus on the subset of the data that share a more recent common ancestry (better approximate transmission chains by subtype); perhaps then the signal for heritability becomes more clear? How would this affect the OU model? I anticipate that this can make the estimates more accurate.

c) analysing the groupM data in a single phylogeny brings about a higher risk of breaking the constant variance assumption underlying BM

Reply: We agree with those valid concerns raised by the reviewer and restricted our main analysis to HIV-1 subtype B as explained in detail in our reply to comment 1. Unfortunately, sample sizes of non-B subtypes were too small to estimate a separate optimum. However, restricting our analysis to HIV-1 subtype B strengthened our results considerably and we wish to thank the reviewer for bringing this up.

4. The authors make a case for the data not always being very informative. Hence, stating that there is 'clear' evidence for a heritability component is over-interpreting the results (end of Discussion section).

Reply: We moderated this statement in line 385 of our discussion, which now reads:

“In our study, we find evidence for a moderate but significant heritability of the HIV-1 reservoir size 1.5 years after initiation of ART and a more tentative heritability of the long-term dynamics of the HIV-1 reservoir decay.”

Minor remarks:

1. introduction line 85: for the sake of completeness, please also cite Vrancken et al, MEE 2015.

Reply: We added this reference in line 85.

2. the partial *pol* and NFL data sets represent a different compromise between evolutionary information (tree accuracy) and sampling density. I may have overlooked it, but I believe that it is good to make this explicit.

Reply: We agree with the reviewer that an explicit clarification will be helpful for the reader. Further, this comment goes along the line of point 9 of reviewer 3. We have thus added the following sentence in the discussion, line 347, clarifying the differences between the datasets:

“In summary, the two datasets analysed (near full-length genome sequences and partial *pol* sequences) represent a different compromise between evolutionary information and sampling density. Since heritability is the property of the population under consideration, no numerical accordance between estimates from the two population could be expected. The advantage of the partial *pol* (Sanger) dataset is a higher sample size and higher sensitivity to obtain sequences also at lower viral loads due to the shorter amplified and sequenced fragment. The advantage of our NGS dataset is that it captures the near full-length HIV-1 genome, thereby allowing for a more exact phylogenetic reconstruction, which is methodologically more correct. However, it is a key strength of this study that different datasets were available, systematically assessed and compared. This may build the basis for further studies assessing the heritability of HIV-1 reservoir phenotypes without having access to different sequencing datasets.”

Reviewer #2

The manuscript entitled "Heritability of the HIV-1 reservoir size and decay under long-term suppressive ART" by Wan et al. studies a very important question in HIV dynamics: what control the reservoir size and decay in patients undergoing suppressive ART; in particular, whether there is measurable heritability in this trait. Since the size and persistence of the reservoir are thought to control prognosis, this is an important topic. Unfortunately, this reviewer is left unconvinced about the conclusions presented from the data and the analysis presented in the manuscript.

1. The central issue is, of course, confounding. Since HIV is transmitted by contact (e.g., needle-sharing or sexual), and human contact is decidedly nonrandom, the recipient of the infection is often more similar to the donor than to a random person. Thus, naive measures of heritability may reflect properties of the transmission network, rather than of the underlying viral biology that this work claims to be characterizing (vide the last sentence of the abstract "Our findings indicate that viral genetic factors contribute to the HIV-1 reservoir size and hence the infecting HIV-1 strain may affect individual patients' hurdle towards a cure of HIV.")

Reply: We thank the reviewer for acknowledging the importance of our study. We appreciate and implemented many points the reviewer raised (see our answers below). However, we must disagree with the concern of confounding for several reasons:

- In principle, for any observational study, reported associations must be interpreted cautiously, because of the potential for unmeasured confounding factors. At the same time, associations found from observational data are an extremely valuable source of evidence and indicative of causal effects. It should be noted that in accordance with this strength, effects found in observational (cohort) studies are considered as the second highest quality of evidence (after randomized controlled trials), when it comes to clinical decisions and guidelines⁶. Since a randomized trial design is not possible for a heritability study, the current design (corrected for the many known and suspected confounders) corresponds to the highest quality of evidence possible for our research question.
- Thus, the main question is not whether the employed study design is appropriate to assess the heritability of HIV-1 reservoir phenotypes, but whether we can adjust for the major potential confounders. In this context, it is a key strength of the current study that it was conducted in the context of the extremely data-rich Swiss HIV Cohort Study (SHCS), allowing to adjust for crucial confounding factors (such as transmission route, the example mentioned). The quality and representativeness of the SHCS was instrumental for identifying key findings regarding human and viral genomics in HIV-1 research⁷⁻¹². It should be noted that based on the SHCS biobank retrospective sequencing was performed extensively to circumvent potential confounding by sampling. Further, this study builds on our previous study¹ of the same research consortium, which identified the main virological, demographic, and clinical determinants of HIV-1 reservoir size and decay in an unprecedented large number of very well characterized participants. Accordingly, we adjusted the current heritability analysis for all covariables identified as predictors of HIV-1 reservoir size or decay, respectively, in this previous study.
- In terms of statistical analyses, the only possible improvements on this issue could be given by performing additional sensitivity analyses in which these confounders are taken into account in alternative ways:
 - (i) We performed a sensitivity analysis correcting for all numerical covariables more thoroughly by using splines (to minimize residual confounding caused by these variables), see *Figure S15* and *S16* below, which did not change our

results. Thus, for considerations of power (degrees of freedom) and simplicity, we kept our main analysis, while adding additional analyses as *Figures S15 and S16* in the Supplementary material.

- (ii) As discussed in the main point 2 of our revisions, we performed a linear regression analysis on our full dataset (to get the best information available in the full population for adjustment; N = 869) and used the residuals thereof for the adjustment for confounders in the mixed-effect models. *Figure S17 and S18* below displays this sensitivity analysis, which indicates only minor differences to our original analysis. Thus, for simplicity, we kept our main analysis, while adding *Figure S17 and S18* in the Supplementary material.

To conclude, while we are very confident that our results are not an artefact of confounding, the reviewers concern highlights that our explanations on confounding in our manuscript merit improvement. We added the following sentence in the Methods section of the manuscript, line 489:

“We performed a sensitivity analysis to relax the linearity assumption made for continuous covariables. In these sensitivity analyses we used polynomial splines¹³ (*bs()* function in the *r* package *splines*) for all numerical variables (**Supplementary Fig S15-16**). Further, we performed a sensitivity analysis using an adjustment based on residuals of the full dataset (N = 869), thereby avoiding the inclusion of many covariables in an already small dataset and using the best information available in the full population for adjustment (**Supplementary Figure S17-18**).”

Figure S15. Adjusted heritability estimates of HIV-1 reservoir size: Sensitivity analysis of adjusting quantitative variables using polynomial splines.

Figure S16. Adjusted heritability estimates of HIV-1 reservoir decay slope: Sensitivity analysis of adjusting quantitative variables using polynomial splines.

Figure S17. Adjusted heritability estimates of HIV-1 reservoir size: Sensitivity analysis of using residuals of a regression analysis on the full dataset (to get the best information available in the full population for adjustment).

Figure S18. Adjusted heritability estimates of HIV-1 reservoir decay slope: Sensitivity analysis of using residuals of a regression analysis on the full dataset (to get the best information available in the full population for adjustment).

2. The authors are, of course, aware of this and try to correct for it by mixed effect models applied to a breakup of the data into transmission clusters; and then controlling for covariates. Not surprisingly, they find that the estimation errors increase as the threshold for transmission cluster is made more stringent (though they do not discuss whether this is roughly the expect sort-sample-size increase), but, problematically, they also find that the covariance-adjusted –point estimate of the heritability of the reservoir size in the NGS analysis tends to zero as the definition is made more stringent. Even more surprisingly, the trend is reversed if we look at the phylogeny from the more conserved Pol region. This calls for a comparison of the two sets of phylogenies (not only of their shapes in terms of distribution of ‘cherries’), something not presented in the manuscript.

Reply: We acknowledge the reviewer’s concern and agree that this merits clarification. Please note, that the specific concern regarding the heritability point estimate of the HIV-1 reservoir size in the NGS analysis was resolved by our new main analysis: when restricting the analysis to subtype B (as suggested by all three reviewers) , the effect does not tend anymore to zero for the strictest threshold (please see response 1 to reviewer 1 and our new *Figures 2 and 3*).

However, the effect the reviewer is referring to (i.e. our previous finding of 0 heritability for the strictest threshold in the adjusted analysis – now shown in *Supplementary Figure S3*) can be explained in terms of the small sample sizes, i.e. small power, of the analyzed clusters for strict thresholds: To show this point, we did a sensitivity analysis, in which we sampled the clusters extracted with the three more liberal thresholds down to the sample size present for the strictest threshold. We found that the cumulative probability of heritability

estimates of zero was very high (30-50% for different thresholds, see our new *Supplementary Figure S19* below).

Figure S19. Cumulative probability of heritability estimates for HIV-1 reservoir size with small sample size using mixed-effect model. Among clusters extracted using each threshold, 9 clusters were randomly selected for 100 times to achieve the cumulative probability. Confidence intervals were calculated respectively from 100 bootstrapped trees.

3. The phylogenetically corrected methods are more interesting once corrected for the covariates. Here one finds that the better determined method that assumes an undirected Brownian evolution finds no effect. A more complex model (that seems to be chosen by AIC and a not-fully-described likelihood ratio method, but not by BIC) seems to find a significant effect (the control for covariates in this method is indirect, though possibly adequate). The problem, however, is that the method may be ill-suited when the sample is mixed for subtype (since it forces a single "optimal" value on all sequences), and in the actual implementation, it encountered flat-likelihood directions, and in some cases the posterior distribution was equal to the prior. In a situation like this, it is not a priori clear whether the results are being forced by the choice of these other priors, or represents a true effect.

Reply: We thank the reviewer for raising this very important point, which goes along several comments of reviewer 1 – please especially consider our reply to comment 1 of reviewer 1. It is possible that different subtypes may have different optima, and that enforcing a global optimum may be wrong. Therefore, we have restricted our main analysis to HIV-1 subtype B, which has substantially strengthened our findings.

Additionally, the reviewer raises the point of smaller estimates from BM compared to OU models. We still find this effect in our new HIV-1 subtype B analysis; however, now the BM

model also finds a significant heritability using the NGS data. Further, it is important to note that it is an established bias of the BM model (which is a special case of the OU model without accounting for stabilizing selection) to underestimate heritability³. We highlighted this bias of the BM model on line 169-170.

4. I have a few minor comments as to presentation. In many places the manuscript reads a bit imprecise: e.g., 'different methods yielded more comparable heritability estimates' (what does "more comparable" mean?) or "Generally, adjusting for potential covariables lowered our heritability estimates in most cases, which indicated that the part of trait variation which was originally attributed to genetic factors when no covariables were specified was actually explained by those chosen covariables." (what does "most cases" mean: this is not what I see in the figures). Also, I do not see that "Overall, we found that viral genetics explained a moderate but significant fraction of the HIV-1 reservoir size variability independent of model choice and robust to the adjustment for covariables." is an accurate description of the state of affairs. At other places, the statements are difficult to parse: "However, heritability estimates larger than zero were found using the genetic information contained in gag and env genes." (how exactly did the genetic information, rather than their variation informing phylogenetic inference, used to deduce heritability?)

Reply: We thank the reviewer for pointing this out. We improved precision of wording in the following places of the manuscript,

- We deleted "different methods yielded more comparable heritability estimates" in line 178. The corresponding sentence now reads:

"Upon adjustment for covariables, heritability estimates derived using near full-length genome sequences from the OU model decreased from 24% to 21%, stayed 10% for the BM model and increased for all mixed-effect model thresholds to values between 29% and 78%. For partial pol sequences estimates decreased from 12% to 7% for the OU model, from 3% to 2% for the BM model, and from 69% to 57% for the strictest ME threshold. For all other ME thresholds, adjusting for covariables increased the heritability estimates to values between 25% and 61% (Fig 2)."

- The sentence in line 289, which previously read "Generally, adjusting for potential covariables lowered our heritability estimates in most cases," was changed to :

"Generally, adjusting for covariables lowered or left unchanged heritability estimates derived with OU and BM model in most cases (except slope heritability estimate derived with BM model using pol sequence) and increased heritability estimates derived with ME model (except for heritability estimates derived with the strictest ME threshold). This confirms the necessity of the adjustment, especially for long-term clinical measures, which were usually influenced by various clinical and host variables."

- In line 200 the sentence "Overall, we found that viral genetics explained a moderate but significant fraction of the HIV-1 reservoir size variability independent of model choice and robust to the adjustment for covariables" was replaced:

"Overall, we found that viral genetics explained a part of the HIV-1 reservoir size variability, which was dependent on heritability estimator and dataset choice, but remained 10% or above for all near-full HIV-1 genome-sequence approaches (Fig 2)".

- In line 229 the sentence "However, heritability estimates larger than zero were found using the genetic information contained in gag and env genes." was replaced:

“However, we found positive heritability estimates using *gag* and *env* sequences. These estimates were above 13% except for the estimate using *gag* sequences under the most liberal mixed-effect model threshold (**Supplementary Fig S7**). “

5. My overall conclusion is that this is data that would be of value to the field, but the claims made in the manuscript seem overly strong.

Reply: We thank the reviewer for noting the potential impact of our study and hope that with the adjustment outlined in response to your concern 4, the claims made in the manuscript do not seem overly strong anymore.

Reviewer #3 (Remarks to the Author):

The understanding of the reservoir size is important to eradication or functional cure of HIV-1 infection. Wan et al studied over 800 treated patients and use the sequences generated by the Sanger or NGS method from plasma samples closest to the initiation of ART to predict the heritability of HIV-1 reservoir size and decay. Both pol and near full-length genomes sequences were studied using three different algorithms, and many factors that can affect the results were adjusted, although how they were adjusted was not clearly stated. They found a moderate but significant fraction of the reservoir size 1.5 years after ART was explained by genetic factors and less significant for the reservoir decay. While the results seems interesting, the study has serious issues with a number of major concerns. The lack of experimental details also make the manuscript very difficult to understand. It is not clear how the results will help to reduce the reservoir size for better clinical outcomes.

Reply: We thank the reviewer for acknowledging that our results are interesting. We would like to highlight that the goal of this study was to investigate if and to what extent viral genetics play a role in the establishment and depletion of the HIV-1 viral reservoir and thereby in HIV-1 pathogenesis. Thus, a detailed explanation of the experimental procedures is beyond the scope of this study and especially given the length of this manuscript and Supplementary material already, we would like refer to our previous study, which explains the experimental procedures in detail¹.

Moreover, while we agree that addressing this question may have no direct clinical impact, we believe that understanding whether the viral genome has an impact on the size and the decay of the HIV-1 reservoir is key for understanding the hurdles towards a potential cure of HIV. For example, our work justifies the future search of individual viral genetic factors steering reservoir dynamics, which may result in important mechanistic insights. Moreover, our results may help to identify potential candidates for future cure strategies as it is assumed that individuals with lower HIV-1 reservoirs and presumably faster decay rates have a higher chance to be cured. More generally, many key observations in viral genetics were made without having any therapeutic effect at the time they were identified. The SHCS has a long-standing activity in this field. For instance, it was instrumental in defining the contribution of the host and viral genome to the setpoint viral load in untreated patients^{7,8,11}. Again, while such studies do not affect HIV clinical care, they are nevertheless important to understand viral pathogenesis and thereby guide future clinical or epidemiological work.

1. The total DNA level is much bigger than the real reservoir size in each patient. It is not the same as real reservoir and not a reliable proxy for the reservoir size.

Reply: We agree with the reviewer that we did not sufficiently discuss this point in the manuscript so far. It is true that by measuring total HIV-1 DNA, we are not distinguishing between replication competent- and -defective viruses. However, in our previous study¹, we found that total HIV-1 DNA measured in PBMC is a robust proxy for the latent HIV-1 reservoir size after the first rapid decay following initiation of ART for several reasons: It correlates independently with time to initiation of ART, with CD4+ cell count, and with viral blips. In addition, the decay of the total HIV-1 reservoir we observed¹ is in line with that observed in smaller studies using either viral outgrowth assays or total HIV-1 DNA^{14,15}. Also other studies have found total HIV-1 DNA in peripheral blood mononuclear cells samples to be a sensitive, clinically relevant marker for the HIV-1 reservoir, which importantly (and in contrast to the viral outgrowth assay) can be measured in larger populations^{16,17}. Levels of total HIV-1 DNA in peripheral blood mononuclear cell were shown to strongly correlate with the levels of inducible virions^{18,19} and viral rebound after treatment interruptions^{20,21}.

So far there is no single perfect measurement for the latent HIV-1 reservoir and this for many reasons: i) anatomically: one will never be able to sample all anatomical and cellular reservoirs at a given time simultaneously, ii) the variety of assays available (e.g. qVOA,

TILDA, PCR based assays for total, integrated and non-integrated proviral DNA, for replication competent proviral DNA) all have their strength and weaknesses, iii) the very recently described, sophisticated methods are not applicable to a cohort of over 1000 participants, as they are too time- and labor intensive and/or require a high amount of cells, and thus, only (prospectively) applicable to selected individuals²²⁻²⁵.

Of note, total HIV-1 DNA assays do not miss parts of the HIV-1 reservoir, because it measures the total sum of all integrated and non-integrated HIV-1 DNA in a given sample. Over the last decades we have learnt that even in patients optimally treated with ART there is still some residual inflammation remaining. Potentially, this inflammation is also due to non-replication competent viruses which, can be transcriptionally active to some extent^{26,27}. Thus, the HIV-reservoir and its different components may have distinct effects on HIV-1 pathogenesis, and thus, a more holistic approach to measure it may be of value. However, we agree with the reviewer that these previous findings should be pointed out in the Discussion and hence we have added the following sentence to line 369:

“In addition, we used total HIV-1 DNA measured in peripheral blood mononuclear cells samples (PBMC) as a proxy for the latent HIV-1 reservoir. While this is theoretically a limitation of our study, it has been shown that total HIV-1 DNA measured in PBMC samples is a sensitive, clinically relevant proxy for the HIV-1 reservoir, which (in contrast to the viral outgrowth assay) can be determined in larger populations needed for heritability studies^{1,16,17}.”

2. The critical conclusions “moderate but significant” and “more tentative” were not well justified. No p values could be found for either conclusion.

Reply: We understand the reviewers’ concern. However, it is agreed that p-values can be misleading and hence instead of relying on p-values, it is recommended to report results using point estimates and their more informative confidence intervals. The advantage of confidence intervals in comparison to p-values is that confidence intervals provide information about statistical significance, as well as the direction and strength of the effect. Thus, we believe that our statements including “moderate but significant” are well justified considering the confidence intervals shown. Note also that our conclusions based on bootstrap-based confidence intervals were broadly in accordance with the fact that we found in most cases significant random effects of ME model estimates (shown in our updated *Table S5* below) found using alternative statistical approaches (likelihood ratio tests, randomization) for HIV-1 reservoir size using Sanger sequences (and thereby being well powered).

Sequences	Method	N	Reservoir size		Reservoir decay slope	
			Unadjusted	Adjusted	Unadjusted	Adjusted
NGS near full-length genome	BM	350	0.000**	0.000**	0.000**	0.000**
	OU	350	0.000**	0.000**	0.000**	0.000**
	ME-0.09	74	0.056	0.048*	0.029*	0.428
	ME-0.06	44	0.253	0.168	0.080	0.096
	ME-0.05	23	0.283	0.181	0.157	0.177
	ME-0.04	8	0.668	1.000	0.020*	0.521
Sanger partial pol	BM	596	0.000**	0.000**	0.000**	0.000**
	OU	596	0.000**	0.000**	0.000**	0.000**
	ME-0.045	138	0.046*	0.057	1.000	1.000
	ME-0.03	85	0.014*	0.005**	1.000	1.000
	ME-0.02	62	0.023*	0.005**	1.000	1.000
	ME-0.01	23	0.006**	0.161	1.000	1.000

Table S5. (HIV-1 subtype B only): *p* value from the significance test of the estimated heritability. For mixed-effect model, the *p* value was obtained with the likelihood ratio test of the random effects (transmission clusters), using *ranova* function from the R package *lmerTest* v 3.1-0²⁸. For BM/OU model, the *p* value was obtained from the likelihood ratio test comparing the fitted model with corresponding white-noise model where all the phenotypic variance was explained by a Gaussian model, using *lrttest* from the R package *lmtree* v 0.9-36. **: *p*<0.01, *: *p*<0.05.

3. It is unclear what phylogenetic information of HIV-1 sequences was used to predict heritability of the reservoir size. Are the lengths of the branches?

Reply: We thank the reviewer for this question. As described in the Methods section (line 416-420) of our manuscript, different information of HIV-1 sequences was used to predict the heritability of the reservoir size and decay: The OU and BM model rely on the evolutionary history given by the phylogeny plus a model of the processes of trait evolution (please find more details in our reply to your question/comment 8 on how heritability estimates were obtained), whose parameters were estimated using the distribution of the considered trait (in this case reservoir size 1.5 years after initiation of ART and reservoir decay 1.5-5.4 years after initiation of ART) on the phylogeny. The ME model only relies on tree topology and within-cluster similarity to estimate heritability. Importantly, the ME model does not assume a model of trait evolution to estimate heritability. The different genetic information which is used by different heritability estimators was actually one of the main reasons to presenting heritability estimates from different models in the underlying study (see also main points 5, 6 and the “General Considerations” in response to comment 1 of reviewer 1).

4. It is likely so since only one sequence was generated by the Sanger method. Since the lengths of infection will have significant effects on branch lengths in phylogenetic trees, the uncertain infection time that were not well defined among most of the cases is a big concern. Will the ambiguous bases in the sequences from chronically infected patients affect the results, especially when a large number of sequences are analyzed together?

Reply: We acknowledge the reviewers' concern and now discuss this (see below). Sanger sequences include ambiguous nucleotide calls, while NGS sequences are obtained by using a majority rule. Even though there is this discrepancy between Sanger and NGS sequences of what the best available information is, this does not affect tree reconstruction, where ambiguous nucleotide calls were treated as missing values. This means that ambiguous positions were not considered for the likelihood calculations and this a common feature of all HIV population-level phylogenies. However, we agree with the reviewer that this was previously not mentioned and merits explanation in our Methods section. We adapted the following sentence in line 432 accordingly:

“Tree construction was performed using the maximum likelihood algorithm RAxML version 8 with the GTRCAT model, treating ambiguous nucleotide calls as missing values”

To further address the reviewers' concern regarding time of infection, we performed a sensitivity analysis, in which we numerically corrected for the estimated date of infection. We used a hierarchical approach to estimate the HIV-1 infection date on the basis of indicators of varying reliability according to Rusert et al.²⁹, as described in our Methods section, line 503. Our sensitivity analysis found almost no differences in the estimated heritability of the HIV-1 reservoir 1.5 years after initiation of ART for the ME estimator. For OU and BM models the corresponding heritability estimates increased slightly when numerically corrected for time since infection, see *Figure* below. This sensitivity analysis could be added to our Supplementary material if requested. Finally, please note, that it is a key strength of our study that the genotype was measured as close as possible to the phenotype (i.e. as close to viral suppression as possible): The samples used for NGS sequencing were chosen intentionally as the last before initiation of ART; similarly, we chose for each patient the Sanger sequence sampled closest to ART initiation.

Figure. Adjusted heritability estimates of HIV-1 reservoir size: sensitivity analysis of adjusting time since infection numerically, compared to the estimates in main analysis where it was adjusted categorically.

5. If the branch lengths are a major factor, will different subtypes have more impact on reservoir sizes; ie different subtypes have huge different in reservoir sizes?

Reply: We want to thank the reviewer for raising this very important point, which was also highlighted by reviewers 1 and 2. In our previous study¹, we found indications that people living with HIV infected with non-B subtypes may indeed have a lower HIV-1 reservoir size. Since this could potentially have disrupted our OU model (assuming a global reservoir optimum), we have adapted the whole manuscript and restricted all our analyses to subtype B. Please consider our response to point 1 of reviewer 1 for a detailed discussion. Since resolving this important issue helped to strengthen our results considerably, we wish to acknowledge the reviewers again for bringing this up.

6. It is unclear how the NGS data were used. Is the consensus sequence were generated for each patients? Again, how the quasispecies populations of chronic viruses were treated?

Reply: The reviewer is correct that the consensus sequence was generated from the NGS sequences to reconstruct the tree. Please note, that this approach is the established standard for inferring the genetic impact on a phenotype at the population level in the field

8,30

However, we agree that details of the diversity of virus populations could provide additional interesting insights, hence we have added the following sentence in our limitations section, line 374:

“Further, we did not analyze the impact of the diversity of viral populations on the HIV-1 reservoir phenotypes, which will merit further investigations in future studies.”

7. The huge variation among three different algorithms and among different thresholds for the ME method is troubling. What is the consensus? Which is more reliable to use?

Reply: We acknowledge the reviewers' concern. However, unfortunately, there is no consensus of cluster thresholds for ME models (and other cluster-based analyses) in the field. The underlying reason for this is the tradeoff between statistical power (higher sample size for liberal thresholds) and methodological correctness (theoretical correctness for strict thresholds) – please note that we also discuss this in point 4 of the main points of our revisions. Thus, we aimed at giving the reader a systematic assessment showing 4 thresholds from very strict to liberal. For the stricter thresholds, statistical power is low and the uncertainty of the estimates accordingly high. When comparing the heritability estimates for different thresholds it is important to take into account the broad confidence intervals for stricter thresholds, which indicate that the higher heritability estimates are not inconsistent with obtained for more liberal thresholds. Finally, it should be noted that (in the revised analysis) the ME analysis yields heritability estimates >20% for all thresholds for HIV-1 reservoir size 1.5 years after ART initiation, indicating a substantial heritability independently of the chosen threshold.

8. It is unclear how the percentages of heritability were obtained. Is 18% too high or 2% too low? Which one is more accurate?

Reply: As summarized in the main points of our revision (points 5 and 6), it is important to note that the different methods each have their strengths and weaknesses and that hence a clearly superior approach cannot be identified. To provide the reader with the full picture of heritability estimators widely used in the field, heritability estimates were obtained using three models: (i) mixed-effect (ME) model, (ii) Brownian motion (BM) model and (iii) Ornstein Uhlenbeck (OU) model.

- (i) For ME models, trait value data was fitted with a mixed-effect model and the heritability was determined as the ratio of the between-cluster variance and the total variance.
- (ii) The BM model assumes a trait evolution along the tree according to BM process, which is a stochastic process σdW_t , where $(W_t)_{(t \geq 0)}$ denotes a BM and accounts for randomness in the divergence of a trait, and σ scales the magnitude of fluctuations.
- (iii) The OU model is an extension of the BM model, adding an additional stabilizing selection towards the long-term optimal trait value. The process can be written as $\alpha(\theta - X)dt + \sigma dW$, where θ denotes the global optimum level and α the strength of selection.

For (i) and (ii) the R package POUMM was used for fitting the OU and BM model with Bayesian inference and maximum likelihood optimization respectively. We reported in the main text the empirical (time-independent) heritability estimates from the package. We additionally adjusted for covariables by doing a multivariate linear regression first to include the influence of covariables and then took the residuals from the regression as trait value input into the optimization.

Regarding the questions of whether 18% is too high or 2% too low, please consider our response to the remark 7 above. Also note that discrepancies decreased considerably in the revised version of the manuscript, thanks to very valuable comments on subtype inclusion made by all the reviewers.

9. Different gene sequences and whole genome sequences have different predicative values. It is not clear which should be more reliable to use.

Reply: We understand the reviewers concern. However, to provide the reader with a full picture, we showed a wide range of analyses on two datasets. The advantage of the partial *pol* (Sanger) dataset is a higher sample size and less bias concerning a generally lower sensitivity for near full-length NGS sequencing (full-length NGS sequences can often not be derived for low virus loads), while the advantage of the near full-length genome (NGS) dataset is that it captures the near full-length genome, thereby allowing for a more exact phylogenetic reconstruction. Thus, methodologically the near full-length genome dataset is closer to the ground truth and more correct, but also potentially underpowered and underrepresenting low-virus-load samples. We integrated a clarification on this to our response to the minor comments 2 of reviewer 1, starting in line 350 and containing the following sentence:

“The advantage of the partial *pol* (Sanger) dataset is a higher sample size and higher sensitivity to obtain a sequence from plasma, while the advantage of our NGS dataset is that it captures the near full-length HIV-1 genome, thereby allows for a more exact phylogenetic reconstruction and is thus methodologically more correct.”

10. Why is the transmission cluster important for the analysis?

Reply: Transmission clusters are only important for the linear ME model, for the following reason: While the classical quantitative genetics approach to estimate heritability is parent-offspring regression, ME models using transmission clusters are an alternative, theoretically identical approach to measure heritability. In ME models the transmission clusters are viewed as independent groups. Using ME model has the advantage of allowing to correct for possible confounders (as done in the underlying study) and also the advantage of allowing to include clusters instead of pairs, which increases the sample size. In summary, the transmission clusters are important because they determine the groups on which the ME models are based. The validity of this approach was tested extensively in one of our recent publication².

11. The actual slope data should be shown for readers to better understand the data for figure 3. 5.4 years do not seem long term.

Reply: We agree with the reviewer that the actual slope data will be helpful for the understanding of readers. The slope data of the 1'057 individuals (see *Figure 1*) is published in full detail in¹. We have now added a Supplementary *Figure S12* depicting the reservoir size 1.5 year after initiation of ART as well as the decay of the reservoir 1.5 to 5.4 years after initiation of ART, for both our populations with near-full genome sequences (population A) and the population with Sanger sequences (population B). We added an according sentence to our Methods section line 413:

“The distributions of HIV-1 reservoir size and decay in Population A and B are displayed in Supplementary Fig S12.”

Figure S12. Violin plots of HIV-1 reservoir size (subplot a) and decay slope (subplot b) in population A and population B. N represents the number of individuals in the two population. The individual observations are shown in small circles and the average is shown as black line.

References

1. Bachmann, N. *et al.* Determinants of HIV-1 reservoir size and long-term dynamics during suppressive ART. *Nat. Commun.* (2019).
2. Bachmann, N. *et al.* Parent-offspring regression to estimate the heritability of an HIV-1 trait in a realistic setup. *Retrovirology* **14**, 33 (2017).
3. Mitov, V. & Stadler, T. A Practical Guide to Estimating the Heritability of Pathogen Traits. *Mol. Biol. Evol.* **35**, 756–772 (2018).
4. Struck, D., Lawyer, G., Ternes, A.-M., Schmit, J.-C. & Bercoff, D. P. COMET: adaptive context-based modeling for ultrafast HIV-1 subtype identification. *Nucleic Acids Res.* **42**, e144–e144 (2014).
5. To, T.-H., Jung, M., Lycett, S. & Gascuel, O. Fast Dating Using Least-Squares Criteria and Algorithms. *Syst. Biol.* **65**, 82–97 (2016).
6. Saag, M. S. *et al.* Antiretroviral Drugs for Treatment and Prevention of HIV Infection in Adults: 2018 Recommendations of the International Antiviral Society–USA Panel Antiretroviral Drugs for Treatment and Prevention of HIV Infection in Adults Antiretroviral Drugs for Treatme. *JAMA* **320**, 379–396 (2018).
7. Alizon, S. *et al.* Phylogenetic approach reveals that virus genotype largely determines HIV set-point viral load. *PLoS Pathog.* **6**, e1001123 (2010).
8. Blanquart, F. *et al.* Viral genetic variation accounts for a third of variability in HIV-1 set-point viral load in Europe. *PLOS Biol.* **15**, e2001855 (2017).
9. Kouyos, R. D. *et al.* Tracing HIV-1 strains that imprint broadly neutralizing antibody responses. *Nature* **561**, 406–410 (2018).
10. Kadelka, C. *et al.* Distinct, IgG1-driven antibody response landscapes demarcate individuals with broadly HIV-1 neutralizing activity. *J. Exp. Med.* **215**, 1589–1608 (2018).
11. Fellay, J. *et al.* A Whole-Genome Association Study of Major Determinants for Host Control of HIV-1. *Science* **317**, 944 LP – 947 (2007).

12. Rauch, A. *et al.* Genetic variation in IL28B is associated with chronic hepatitis C and treatment failure: a genome-wide association study. *Gastroenterology* **138**, 1338–45, 1345.e1–7 (2010).
13. R Core Team. R: A Language and Environment for Statistical Computing. R Foundation for Statistical Computing. Vienna, Austria. (2019).
14. Finzi, D. *et al.* Identification of a Reservoir for HIV-1 in Patients on Highly Active Antiretroviral Therapy. *Science* **278**, 1295 LP – 1300 (1997).
15. Siliciano, J. D. *et al.* Long-term follow-up studies confirm the stability of the latent reservoir for HIV-1 in resting CD4+ T cells. *Nat. Med.* **9**, 727–728 (2003).
16. Avettand-Fenoel, V. *et al.* Total HIV-1 DNA, a Marker of Viral Reservoir Dynamics with Clinical Implications. *Clin. Microbiol. Rev.* **29**, 859–880 (2016).
17. Anderson, E. M. & Maldarelli, F. The role of integration and clonal expansion in HIV infection: live long and prosper. *Retrovirology* **15**, 71 (2018).
18. Kiselina, M. *et al.* Integrated and Total HIV-1 DNA Predict Ex Vivo Viral Outgrowth. *PLoS Pathog.* **12**, e1005472 (2016).
19. Cillo, A. R. *et al.* Blood biomarkers of expressed and inducible HIV-1. *AIDS Lond. Engl.* **32**, 699–708 (2018).
20. Yerly, S. *et al.* Proviral HIV-DNA predicts viral rebound and viral setpoint after structured treatment interruptions. *AIDS Lond. Engl.* **18**, 1951–1953 (2004).
21. Williams, J. P. *et al.* HIV-1 DNA predicts disease progression and post-treatment virological control. *eLife* **3**, e03821 (2014).
22. Gaebler, C. *et al.* Combination of quadruplex qPCR and next-generation sequencing for qualitative and quantitative analysis of the HIV-1 latent reservoir. *J. Exp. Med.* **216**, 2253 LP – 2264 (2019).
23. Bruner, K. M. *et al.* A quantitative approach for measuring the reservoir of latent HIV-1 proviruses. *Nature* **566**, 120–125 (2019).
24. Cohn, L. B. *et al.* Clonal CD4(+) T cells in the HIV-1 latent reservoir display a distinct gene profile upon reactivation. *Nat. Med.* **24**, 604–609 (2018).
25. Einkauf, K. B. *et al.* Intact HIV-1 proviruses accumulate at distinct chromosomal positions during prolonged antiretroviral therapy. *J. Clin. Invest.* **129**, 988–998 (2019).
26. Fischer, M. *et al.* Biphasic decay kinetics suggest progressive slowing in turnover of latently HIV-1 infected cells during antiretroviral therapy. *Retrovirology* **5**, 107 (2008).
27. Yukl, S. A. *et al.* HIV latency in isolated patient CD4⁺ T cells may be due to blocks in HIV transcriptional elongation, completion, and splicing. *Sci. Transl. Med.* **10**, eaap9927 (2018).
28. Kuznetsova, A., Brockhoff, P. B. & Christensen, R. H. B. lmerTest Package: Tests in Linear Mixed Effects Models. *J. Stat. Softw.* **82**, 1–26 (2017).
29. Rusert, P. *et al.* Determinants of HIV-1 broadly neutralizing antibody induction. *Nat. Med.* **22**, 1260–1267 (2016).
30. Bertels, F. *et al.* Dissecting HIV Virulence: Heritability of Setpoint Viral Load, CD4+ T-Cell Decline, and Per-Parasite Pathogenicity. *Mol. Biol. Evol.* **35**, 27–37 (2018).

Reviewers' Comments:

Reviewer #1:

Remarks to the Author:

I only have a few questions left.

1) Concerning the rooting with LSD, the authors should provide the estimated rate of evolution as well as the tMRCA for each of the data sets as these are good indicators for the presence of significant time signal. I inquire for this because in the absence of significant time signal, the direction of evolution will be unreliable estimated, and this obviously can impact results.

A few additional remarks have come to mind when reading the revised version of the manuscript:

1) line 519: "Time to viral suppression was defined as the time from initiation of ART until the first viral load below 50 copies/ml HIV-1 plasma RNA. If this was not reached, 5.4 years was set as time to viral suppression."

=> How does this assumption impact the heritability of the decay rate?

2) table s4: it is bizarre that the loglik (decay slope) is the same for the null and ME models for the pol-dataset. Is this because of a an oversight?

5) line 331: incorporating measurement error is one of the strengths of a fully Bayesian approach to this problem. If the authors could indicate this in the discussion, including appropriate reference(s)?

6) a reply on a remark given in response to comment 4 of Reviewer #3. The authors state that "... this [ignoring ambiguity base calls] is a common feature of all HIV population-level phylogenies.". => This is not correct. Optimal phylogenetic resolution is achieved by taking the information of ambiguity base calls into account. As there is no good reason for not doing so, this permeates the field. Whether or not this will substantially impact the results, however, depends on their abundance.

7) figs s2 and s7 are difficult to interpret: please rework such that the heritability estimates are potted next to each other as in e.g. bee swarm plots.

Finally, the manuscript will improve with more attention to the language. For example: line 305, 'which confirms': this is not a precise expression of the implication of the similar-sized heritability between spVL and RNApreART. Another example of imprecise wording is at line 215: 'were larger than zero': this is a statement that cannot be made as the confidence intervals include zero. For the same reason, please also update the description of results for fig s7. There are other example that I do not list here - please carefully revise language throughout the manuscript.

Reviewer #2:

Remarks to the Author:

The new manuscript has addressed the concerns I raised before. I appreciate the reanalysis and the careful reworking by the authors: I no longer think that the claims are overly strong.

A general remark about the issue of observational studies that the authors bring up in the rebuttal: I agree they are useful and the primary source for raising hypotheses, and, in fact, I posit further that we will increasingly need to extract information from such studies. The issue I tried to raise, not very articulately I fear, is that they need care in interpretation: in particular, if one sees a *reduction* in

the effect as one makes the estimates more unbiased (e.g., by a stricter threshold in mixed effects model), then one should be concerned. The new results from the B subtype alone allay the main thrust of my concern.

Near line 525. Worth making two points

- 1) Adjustment for covariates almost never changed the estimates outside the 95% CI; though the adjusted values have larger errors as expected from a more complex model.
- 2) Almost all the determinations other than BM are consistent.

Similar comment near line 647.

lines 760: I got confused by the discussion. Are the authors suggesting that the differences may be due to the lack of resolution in the phylogenetic reconstruction from the conserved pol region? I think that is a reasonable argument, but could be made more clearly. (Another point about the exposition: the issue is less that one gets very small central estimate of the heritability, rather it is that the upper limit of the CI is too low to be consistent with the NGS analysis!)

line 904: I do not think that the case has been made for the *necessity* of the adjustment since the adjusted and unadjusted confidence intervals overlap so well. It is also difficult to argue this from many of the adjustment effects going in the same direction since the different measures are obviously not uncorrelated. I would make a weaker statement along the lines: "The inclusion of the covariates leads to more complex models and resultingly higher uncertainties, but reduce the unquantified bias in the analyses."

line 1039: it is not the heritability that is tentative, but the evidence for it. (Same statement in the abstract.)

Tanmoy Bhattacharya

Reviewer #3:

Remarks to the Author:

This reviewer appreciate the effort that authors put into their responses to address concerns from all reviewers, especially analyzing the subtype B sequences separately to avoid the unpredictable effects of different subtypes. Unfortunately, some my major concerns are not fully addressed.

1. The total DNA level simply cannot be used to as a reliable proxy for the reservoir size. The leading experts, Drs. Siliciano, Coffin and others, have made this very clear. The reason is that the total DNA level is way too bigger (the vast majority of them are from defective genomes) than the real reservoir size in a patient. Thus, the total DNA amount in a sample can completely overshadow the size of the reservoir. However, it is ok to use total DNA levels to measure the DNA loads in the samples.
2. If the branch lengths in phylogenetic trees are used in all models, determination of infection time will be critical. Generally, the HIV-1 accumulated 1% diversity per year. Comparing the viruses from 1-year infection to those from 10-year infection will be a very different thing. The authors stated that the infection time was corrected. But it is not clear how it was corrected and based on what.
3. As a virologist, I still don't understand how the models work. The authors should explain this to the general readers better. Are all variants equally important to the heritability? If so, how can synonymous mutations contribute? How APOBEC mediated hypermutations that is very frequent in proviral genome sequences affect the models? Can heritability be determined by some particular mutations? If so, can this be identified.
4. While it is understandable that the authors tried to present many different models and use different

viral regions to present as many scenarios as possible. But the high variable results among them will make the readers completely puzzled about what should be the best to use.

Point-by-point reply (round 2)

We are grateful for the reviewers' positive assessment of the revised version of our manuscript and their appreciation of the major changes done in these revisions. We would also like to thank the reviewers for their additional comments, which have been addressed in this point-by-point reply. Our point-by-point reply contains **reviewer's comments in blue**, our answers in black, and **changes in the manuscript in brown**.

Reviewer #1 (Remarks to the Author):

I only have a few questions left.

Reply: We would like to thank the reviewer for the positive evaluation of our revised manuscript.

1) Concerning the rooting with LSD, the authors should provide the estimated rate of evolution as well as the tMRCA for each of the data sets as these are good indicators for the presence of significant time signal. I inquire for this because in the absence of significant time signal, the direction of evolution will be unreliable estimated, and this obviously can impact results.

Reply: We would like to thank the reviewer for indicating this very relevant point. We now provide the estimated rate of evolution as well as the tMRCA for each data set in our **Supplementary table S10** and indicate this in the legends of our Figure S13 and S14: "The rate of evolution and tMRCA estimated by LSD can be found in Table S10."

	Estimated rate of evolution (substitutions per site per year)	tMRCA (calendar year)
NGS near full-length genome	0.003439 [0.002849, 0.003886]	1953 [1944, 1962]
Sanger partial pol genome	0.003198 [0.002088, 0.003306]	1967 [1942, 1968]

Table S10. Estimated rate of evolution and tMRCA with the HIV-1 NGS near full-length genome and Sanger partial *pol* genome dataset, using LSD-0.2¹ (both based on HIV-1 subtype B only). 95% confidence intervals are shown in square brackets. While the estimated tMRCA is earlier when based on near full viral genomes than when based on *pol*, it should be noted that the confidence intervals are broad and overlapping.

A few additional remarks have come to mind when reading the revised version of the manuscript:

1) line 519: "Time to viral suppression was defined as the time from initiation of ART until the first viral load below 50 copies/ml HIV-1 plasma RNA. If this was not reached, 5.4 years was set as time to viral suppression."
=> How does this assumption impact the heritability of the decay rate?

Reply: We would like to thank the reviewer for pointing out this artifact from an old version of our analysis. All patients had a viral load below 50 copies/ml HIV-1 plasma RNA during the study period and we thus deleted the sentence “*If this was not reached, 5.4 years was set as time to viral suppression*” (as this rule is not applied in the data set used for our analysis).

2) table s4: it is bizarre that the loglik (decay slope) is the same for the null and ME models for the pol-dataset. Is this because of an oversight?

Reply: We would like to thank the reviewer for pointing it out. We double checked this table and found that the fact that we obtained the same loglik (for the DNA decay slope analysis) for the null and mixed-effect models was due to the negligibly small random effects in the latter. i.e., we found virtually a zero heritability for the HIV-1 reservoir decay slope using partial *pol* sequences (Figure 3) and if a variance component is zero, dropping it from the model would not influence the likelihood (see <http://bbolker.github.io/mixedmodels-misc/glmmFAQ.html#zero-variance> and Pasch et al. ² for an explanation of why this can be expected to occur).

Further, when calculating model statistics in Table S4, we applied mixed-effect models to the full dataset to make it comparable to other models in terms of sample size (see Supplementary Method). Thus, the small random effects become further negligible, making mixed-effect model almost the same to the null model.

5) line 331: incorporating measurement error is one of the strengths of a fully Bayesian approach to this problem. If the authors could indicate this in the discussion, including appropriate reference(s)?

Reply: We acknowledge this interesting suggestion. However, an exact quantification of measurement error is not possible in the context of this study. Specifically, we cannot distinguish between measurement error and environmental noise or random variation of the phenotype under consideration. Please note, that our approach is conservative, in the sense that we assume that all variation is variability of the phenotype, which reduces our heritability estimates.

6) a reply on a remark given in response to comment 4 of Reviewer #3. The authors state that “... this [ignoring ambiguity base calls] is a common feature of all HIV population-level phylogenies.”.

=> This is not correct. Optimal phylogenetic resolution is achieved by taking the information of ambiguity base calls into account. As there is no good reason for not doing so, this permeates the field. Whether or not this will substantially impact the results, however, depends on their abundance.

Reply: We thank the reviewer for pointing this out. We actually found out that this was a misunderstanding from our side and that the method used for phylogenetic reconstruction (RaXml) does take into account ambiguities.

7) figs s2 and s7 are difficult to interpret: please rework such that the heritability estimates are potted next to each other as in e.g. bee swarm plots.

Reply: Thanks for raising this point. We improved the two plots using boxplots and replaced them in the Supplementary material.

Figure S2. (HIV-1 subtype B only, **replaced in Supplementary material**): Adjusted heritability estimates of HIV-1 reservoir size across the genome. Heritability was inferred with the mixed-effect model using NGS sequences of population A. The horizontal dashed lines are the viral whole-genome heritability estimates with the same method and thresholds. Small, medium and large thresholds refer to the D2-D4 phylogenetic distance thresholds (Table S1). Point estimates are shown in black dots. Boxplots represent the medium, 25% and 75% quantiles of the 100 bootstrapped estimates. Whiskers represent the 95% confidence intervals.

Figure S7. (HIV-1 subtype B only, **replaced in Supplementary material**): Adjusted heritability estimates of HIV-1 reservoir decay slope across the genome. Heritability was inferred with the mixed-effect model using NGS sequences of population A. The horizontal dashed lines are the whole-genome heritability estimates with the same method and thresholds. Small, medium and large thresholds refer to the D2-D4 phylogenetic distance thresholds (Table S1). Point estimates are shown in black dots. Boxplots represent the medium, 25% and 75% quantiles of the 100 bootstrapped estimates. Whiskers represent the 95% confidence intervals.

Finally, the manuscript will improve with more attention to the language. For example: line 305, 'which confirms': this is not a precise expression of the implication of the similar-sized

heritability between spVL and RNApreART. Another example of imprecise wording is at line 215: 'were larger than zero': this is a statement that cannot be made as the confidence intervals include zero. For the same reason, please also update the description of results for fig s7. There are other example that I do not list here - please carefully revise language throughout the manuscript.

Reply: We improved the wording of the highlighted sentences. These now read:

- line 301: “However, as the heritability estimate for RNApreART in our dataset was 20.9%, and thus comparable to that of SPVL, we believe RNApreART can be used as an approximation for SPVL as a potential covariable.”

- line 211: “Using viral near full-length genome NGS sequences, we found tentative evidence that the HIV-1 reservoir decay slope was heritable in our study population A (Fig 3) with estimates ranging from 3% to 85%.”

Further we carefully re-read the manuscript and improved language in several places.

Reviewer #2 (Remarks to the Author):

The new manuscript has addressed the concerns I raised before. I appreciate the reanalysis and the careful reworking by the authors: I no longer think that the claims are overly strong.

A general remark about the issue of observational studies that the authors bring up in the rebuttal: I agree they are useful and the primary source for raising hypotheses, and, in fact, I posit further that we will increasingly need to extract information from such studies. The issue I tried to raise, not very articulately I fear, is that they need care in interpretation: in particular, if one sees a *reduction* in the effect as one makes the estimates more unbiased (e.g., by a stricter threshold in mixed effects model), then one should be concerned. The new results from the B subtype alone allay the main thrust of my concern.

Reply: We highly appreciate the reviewers' positive assessment of the improvements in our manuscript.

Near line 525. Worth making two points

- 1) Adjustment for covariates almost never changed the estimates outside the 95% CI; though the adjusted values have larger errors as expected from a more complex model.
 - 2) Almost all the determinations other than BM are consistent.
- Similar comment near line 647.

Reply: We are grateful that the reviewer points this out. Regarding the first point, we added these points in the manuscript:

1. Line 178: "Generally, while broader confidence intervals for heritability were obtained in the adjusted models, adjusted and unadjusted estimates were qualitatively consistent for HIV-1 reservoir size. "
2. Line 216: "Similar to heritability estimates for HIV-1 reservoir size, adjustment for covariables increased the confidence intervals while yielding estimates comparable to the unadjusted values."

Regarding the second point, we have added a sentence at line 198 to indicate the consistency of estimates based on different models:

"Overall, we found that viral genetics explained a part of the HIV-1 reservoir size variability, which was dependent on heritability estimator and dataset choice, but remained at 10% or above for all near-full HIV-1 genome-sequence approaches (Fig 2)."

lines 760: I got confused by the discussion. Are the authors suggesting that the differences may be due to the lack of resolution in the phylogenetic reconstruction from the conserved pol region? I think that is a reasonable argument, but could be made more clearly. (Another point about the exposition: the issue is less that one gets very small central estimate of the heritability, rather it is that the upper limit of the CI is too low to be consistent with the NGS analysis!)

Reply: We thank the reviewer for making this very interesting point. There might be a misunderstanding here. In the goodness of fit section (from line 233), we discussed the statistical comparisons between different methods rather than comparisons between different

sequences as you mentioned. But we did explain the difference between estimates using partial *pol* and near full-genome sequences in discussion part (at line 330). There, we pointed out that using different sequences represented a compromise between evolutionary information and sampling density.

We agree that for estimates of decay slope, the upper limit of the confidence interval using partial *pol* sequences is too low to be consistent with the NGS analysis. We discussed this issue at line 223. There, we further compared the heritability estimates using phylogenies built from different genomic regions using near full-length NGS sequences (Figure S7). The estimates using NGS partial *pol* sequences confirmed the qualitative difference between heritability estimates for decay slope using partial *pol* and near full-length genome sequences.

line 904: I do not think that the case has been made for the *necessity* of the adjustment since the adjusted and unadjusted confidence intervals overlap so well. It is also difficult to argue this from many of the adjustment effects going in the same direction since the different measures are obviously not uncorrelated. I would make a weaker statement along the lines: "The inclusion of the covariates leads to more complex models and resultingly higher uncertainties, but reduce the unquantified bias in the analyses."

Reply: We acknowledge the reviewers' comment and have adapted our statement in line 285 accordingly. It now reads:

«Generally, adjusting for covariables lowered or left unchanged heritability estimates derived with OU and BM model in most cases (except the slope heritability estimate derived with the BM model using *pol* sequences) and increased heritability estimates derived with the ME model (except for heritability estimates for the strictest ME threshold). This indicates ~~confirms~~ that the inclusion of the covariables led to more complex models resulting in higher uncertainties, but reduced potential confounding by other covariables such as pre-treatment virus load which is known to affect both the HIV-1 reservoir and to cluster on the viral phylogeny. especially for long-term clinical measures, which were usually influenced by various clinical and host variables.»

line 1039: it is not the heritability that is tentative, but the evidence for it. (Same statement in the abstract.)

Reply: We thank the reviewer for pointing this out. We have adapted the manuscript at the following positions:

3. Line 47: "At the same time, we found a more tentative evidence of the heritability of the long-term HIV-1 reservoir decay."
4. Line 384: "In our study, we find evidence for a moderate but significant heritability of the HIV-1 reservoir size 1.5 years after initiation of ART and a more tentative evidence of the heritability of the long-term dynamics of the HIV-1 reservoir decay.»

Reviewer #3 (Remarks to the Author):

This reviewer appreciate the effort that authors put into their responses to address concerns from all reviewers, especially analyzing the subtype B sequences separately to avoid the unpredictable effects of different subtypes. Unfortunately, some my major concerns are not fully addressed.

1. The total DNA level simply cannot be used to as a reliable proxy for the reservoir size. The leading experts, Drs. Siliciano, Coffin and others, have made this very clear. The reason is that the total DNA level is way too bigger (the vast majority of them are from defective genomes) than the real reservoir size in a patient. Thus, the total DNA amount in a sample can completely overshadow the size of the reservoir. However, it is ok to use total DNA levels to measure the DNA loads in the samples.

Reply: The reviewers' concern points to an ongoing discussion in the field: While we of course agree that total HIV-1 DNA levels are higher than the HIV-1 reservoir of replication competent latently infected cells, we must disagree that total HIV-1 DNA cannot be used as a proxy for the HIV-1 reservoir. The reasons for this have been outlined in detail in the first round of our revisions and were integrated in the discussion section of the manuscript in line 367. Especially the following points should be noted:

In our previous study³, we found that total HIV-1 DNA measured in PBMC is a robust proxy for the latent HIV-1 reservoir size after the first rapid decay following initiation of ART for several reasons: It correlates independently with time to initiation of ART, with CD4+ cell count, and with viral blips. In addition, the decay of the total HIV-1 reservoir we observed³ was consistent with that observed in smaller studies using either viral outgrowth assays or total HIV-1 DNA (published by the leading experts mentioned by the reviewer)^{4,5}. In particular we found similar decay dynamics and determinants to those studies. Also other studies have found total HIV-1 DNA in peripheral blood mononuclear cells samples to be a sensitive, clinically relevant marker for the HIV-1 reservoir, which in contrast to the viral outgrowth assay can be measured in larger populations^{6,7}.

Furthermore, regarding the defective genomes, a recent study by Anthony Fauci's group highlighted that defective proviruses are not silent and are capable of transcribing novel unspliced forms of HIV-RNA transcripts during cART⁸. This phenomenon may further help to demonstrate the biological relevance of total HIV-1 DNA, i. e., that not just replication competent DNA genomes are of biological relevance but also defective proviruses. Such findings may potentially also be relevant for residual inflammation that is found in patients with successfully suppressed viremia.

More generally, it is important to note that so far there is no single perfect measurement for the latent HIV-1 reservoir. This is the case for many reasons: i) anatomically: one will never be able to sample all anatomical and cellular reservoirs at a given timepoint simultaneously in people living with HIV and being alive, ii) the variety of assays available (e.g. qVOA, TILDA, PCR based assays for total, integrated and non-integrated proviral DNA, for replication competent proviral DNA) all have their strength and weaknesses, iii) the very recently described, sophisticated methods are not applicable to a cohort of over 1000 participants, as they are too time- and labor intensive and/or require a high amount of cells, and thus, are only (prospectively) applicable to selected individuals^{9,10}. Such studies of a few individuals, although highly interesting and important, always bear the risk of strong

selection implying that generalizability to the population level is often problematic. Thus, there is a trade-off between the potential bias directly induced by the measurement method and the bias indirectly induced by method's restrictions and limitations of the study population. In this context, we believe that our choice is a good compromise between these two conflicting requirements (and in fact we believe that it is at this time the only choice that allows to estimate viral heritability). However, we agree that at the end of the day both types of studies are needed in an iterative way to move the field forward.

Finally, a new, very recently online published paper by Papasaavas et al in "Clinical Infectious Diseases" shows a significant correlation between total proviral DNA and the intact provirus DNA assay, underlining the utility of total proviral DNA as a proxy for the viral reservoir¹¹.

2. If the branch lengths in phylogenetic trees are used in all models, determination of infection time will be critical. Generally, the HIV-1 accumulated 1% diversity per year. Comparing the viruses from 1-year infection to those from 10-year infection will be a very different thing. The authors stated that the infection time was corrected. But it is not clear how it was corrected and based on what.

Reply: The time since HIV-1 infection date was estimated based on indicators as described in our methods section. Originally, we determined the initiation of ART in acute/chronic infection phase using estimated infection date and adjusted it using categorical variables. Following this reviewers' previous comment 4, we performed and showed in the first round of revisions a sensitivity analysis of adjusting time since infection numerically, which did not disrupt our heritability estimates. Further, please note that time of infection and within-patient evolution become less relevant for liberal thresholds, which argues for showing a variety of models (see your concern 4). Finally, please note that HIV-1 accumulates less than 1% diversity per year^{12,13}.

Also, we made the following changes in the manuscript to further clarify the adjustment for infection time:

Line 502: "We determined the initiation of ART as occurring in acute/chronic infection using the estimated HIV-1 infection date as specified in the previous study³."

3. As a virologist, I still don't understand how the models work. The authors should explain this to the general readers better. Are all variants equally important to the heritability? If so, how can synonymous mutations contribute? How APOBEC mediated hypermutations that is very frequent in proviral genome sequences affect the models? Can heritability be determined by some particular mutations? If so, can this be identified.

Reply: This might be a misunderstanding. In both datasets, we sequenced plasma viruses. Thus, this has nothing to do with proviruses. We further clarified this in line 399:

"5) either A. NGS sequencing of the full genome from plasma at the latest sample before the initiation of ART or B. Sanger sequencing of partial pol region obtained from plasma for genotypic resistance testing (GRT)"

Further, please note that phylogenetic models generally do not treat different variants of the HIV-1 genome differently and that our model compares if viruses with a similar genome have similar phenotypes (which is how heritability is defined).

4. While it is understandable that the authors tried to present many different models and use different viral regions to present as many scenarios as possible. But the high variable results among them will make the readers completely puzzled about what should be the best to use.

Reply: We acknowledge the reviewers' concern. However, the aim of this paper is not to identify the "best" method to measure heritability but rather to provide the reader with a full and systematic assessment of heritability based on the available methodology. Please further note, that no single "best" heritability estimator exists¹⁴ (main point 6 of our reply in the first round of revisions), but that among the heritability estimators each method has advantages and disadvantages. Finally, we strongly believe that it is a key strength of this study that it provides a systematic assessment of heritability estimates for HIV-1 reservoir size and decay dynamics by applying different heritability estimation methods to distinct datasets (thereby showing the true uncertainty of these estimates). This systematic nature of the assessment strengthens our conclusions by showing a consistent pattern of a substantial heritability of the HIV-1 reservoir size but weaker patterns for the HIV-1 decay slope.

References

1. To, T.-H., Jung, M., Lycett, S. & Gascuel, O. Fast Dating Using Least-Squares Criteria and Algorithms. *Syst. Biol.* **65**, 82–97 (2016).
2. Pasch, B., Bolker, B. M. & Phelps, S. M. Interspecific Dominance Via Vocal Interactions Mediates Altitudinal Zonation in Neotropical Singing Mice. *Am. Nat.* **182**, E161–E173 (2013).
3. Bachmann, N. *et al.* Determinants of HIV-1 reservoir size and long-term dynamics during suppressive ART. *Nat. Commun.* (2019).
4. Finzi, D. *et al.* Identification of a Reservoir for HIV-1 in Patients on Highly Active Antiretroviral Therapy. *Science* **278**, 1295 LP – 1300 (1997).
5. Siliciano, J. D. *et al.* Long-term follow-up studies confirm the stability of the latent reservoir for HIV-1 in resting CD4+ T cells. *Nat. Med.* **9**, 727–728 (2003).
6. Avettand-Fenoel, V. *et al.* Total HIV-1 DNA, a Marker of Viral Reservoir Dynamics with Clinical Implications. *Clin. Microbiol. Rev.* **29**, 859–880 (2016).

7. Anderson, E. M. & Maldarelli, F. The role of integration and clonal expansion in HIV infection: live long and prosper. *Retrovirology* **15**, 71 (2018).
8. Imamichi, H. *et al.* Defective HIV-1 proviruses produce viral proteins. *Proc. Natl. Acad. Sci.* **117**, 3704 (2020).
9. Bruner, K. M. *et al.* A quantitative approach for measuring the reservoir of latent HIV-1 proviruses. *Nature* **566**, 120–125 (2019).
10. Cohn, L. B. *et al.* Clonal CD4+ T cells in the HIV-1 latent reservoir display a distinct gene profile upon reactivation. *Nat. Med.* **24**, 604–609 (2018).
11. Papasavvas, E. *et al.* Intact HIV reservoir estimated by the intact proviral DNA assay correlates with levels of total and integrated DNA in the blood during suppressive antiretroviral therapy. *Clin. Infect. Dis.* (2020) doi:10.1093/cid/ciaa809.
12. Shankarappa, R. *et al.* Consistent viral evolutionary changes associated with the progression of human immunodeficiency virus type 1 infection. *J. Virol.* **73**, 10489–10502 (1999).
13. Carlisle, L. A. *et al.* Viral Diversity Based on Next-Generation Sequencing of HIV-1 Provides Precise Estimates of Infection Recency and Time Since Infection. *J. Infect. Dis.* **220**, 254–265 (2019).
14. Fraser, C. *et al.* Virulence and Pathogenesis of HIV-1 Infection: An Evolutionary Perspective. *Science* **343**, 1243727 (2014).

Reviewers' Comments:

Reviewer #4:

Remarks to the Author:

Dear Mrs Schmid,

Dear Mr Rosch,

Concerning the manuscript from Wan and Bachmann entitled: Heritability of the HIV-reservoir size and decay under long- term suppressive ART. The key questions for me have indeed been formulated by Mrs Schmid:

In short, the paper has already been through two rounds of review. Reviewer #1 and #2 who have expertise with virus evolution and heritability had only minor remaining concerns that authors have addressed appropriately. However, reviewer #3 who has expertise with HIV reservoir had major remaining concerns on study design, more specifically on the use of HIV DNA as proxy for the reservoir size.

We feel that the authors and reviewer #3 are at an impasse and are seeking the opinion on this issue from an additional independent scientist. We are hoping that you will be willing to comment on the concerns raised by reviewer #3 and let us know whether in your opinion the authors have appropriately addressed these concerns and whether in your opinion the manuscript is suitable for publication in Nature Communications.

I therefore focused on the specific questions and the concerns raised by reviewer #3 and give my overall thoughts at the end of this document.

Reviewer #3 (Remarks to the Author):

This reviewer appreciates the effort that authors put into their responses to address concerns from all reviewers, especially analyzing the subtype B sequences separately to avoid the unpredictable effects of different subtypes. Unfortunately, some of my major concerns are not fully addressed.

1. The total DNA level simply cannot be used as a reliable proxy for the reservoir size. The leading experts, Drs. Siliciano, Coffin and others, have made this very clear. The reason is that the total DNA level is way too bigger (the vast majority of them are from defective genomes) than the real reservoir size in a patient. Thus, the total DNA amount in a sample can completely overshadow the size of the reservoir. However, it is ok to use total DNA levels to measure the DNA loads in the samples.

Reply: The reviewers' concern points to an ongoing discussion in the field: While we of course agree that total HIV-1 DNA levels are higher than the HIV-1 reservoir of replication competent latently infected cells, we must disagree that total HIV-1 DNA cannot be used as a proxy for the HIV-1 reservoir. The reasons for this have been outlined in detail in the first round of our revisions and were integrated in the discussion section of the manuscript in line 367. Especially the following points should be noted:

In our previous study³, we found that total HIV-1 DNA measured in PBMC is a robust proxy for the latent HIV-1 reservoir size after the first rapid decay following initiation of ART for several reasons: It correlates independently with time to initiation of ART, with CD4+ cell count, and with viral blips. In addition, the decay of the total HIV-1 reservoir we observed³ was consistent with that observed in smaller studies using either viral outgrowth assays or total HIV-1 DNA (published by the leading experts mentioned by the reviewer)^{4,5}. In particular we found similar decay dynamics and determinants to those studies. Also other studies have found total HIV-1 DNA in peripheral blood mononuclear cells samples to be a sensitive, clinically relevant marker for the HIV-1 reservoir, which in contrast to the viral outgrowth assay can be measured in larger populations^{6,7}.

Furthermore, regarding the defective genomes, a recent study by Anthony Fauci's group highlighted that defective proviruses are not silent and are capable of transcribing novel unspliced forms of HIV-

RNA transcripts during cART8. This phenomenon may further help to demonstrate the biological relevance of total HIV-1 DNA, i. e., that not just replication competent DNA genomes are of biological relevance but also defective proviruses. Such findings may potentially also be relevant for residual inflammation that is found in patients with successfully suppressed viremia.

More generally, it is important to note that so far there is no single perfect measurement for the latent HIV-1 reservoir. This is the case for many reasons: i) anatomically: one will never be able to sample all anatomical and cellular reservoirs at a given timepoint simultaneously in people living with HIV and being alive, ii) the variety of assays available (e.g. qVOA, TILDA, PCR based assays for total, integrated and non-integrated proviral DNA, for replication competent proviral DNA) all have their strength and weaknesses, iii) the very recently described, sophisticated methods are not applicable to a cohort of over 1000 participants, as they are too time- and labor intensive and/or require a high amount of cells, and thus, are only (prospectively) applicable to selected individuals^{9,10}. Such studies of a few individuals, although highly interesting and important, always bear the risk of strong selection implying that generalizability to the population level is often problematic. Thus, there is a trade-off between the potential bias directly induced by the measurement method and the bias indirectly induced by method's restrictions and limitations of the study population. In this context, we believe that our choice is a good compromise between these two conflicting requirements (and in fact we believe that it is at this time the only choice that allows to estimate viral heritability). However, we agree that at the end of the day both types of studies are needed in an iterative way to move the field forward.

Finally, a new, very recently online published paper by Papasaavas et al in "Clinical Infectious Diseases" shows a significant correlation between total proviral DNA and the intact provirus DNA assay, underlining the utility of total proviral DNA as a proxy for the viral reservoir¹¹.

Comment Reviewer #4:

Total HIV DNA is related to virological failure in monotherapy studies, both for PI's and for INI's, so there is a link with total HIV DNA and functional reservoir in terms of generation of new viral particles. Reviewer 3 is right that the functional reservoir is much more complex and the field has moved towards the idea that total DNA HIV is not a reliable proxy that can be used in cure studies. Both the intactness of the sequence, the integration site and the chromosomal structure will influence potential transcriptional activity and the formation of new infectious particles. However, for large scale studies we are currently limited by the enormous cost and the lack of high throughput assays to measure those parameters. The authors have outlined that very nicely in the rebuttal letter but only to a limited extend in the manuscript itself. Therefore this should be adapted and integration of the key points raised in the rebuttal letter should be integrated in the manuscript discussion, as this is indeed a major limitation of the study.

2. If the branch lengths in phylogenetic trees are used in all models, determination of infection time will be critical. Generally, the HIV-1 accumulated 1% diversity per year. Comparing the viruses from 1-year infection to those from 10-year infection will be a very different thing. The authors stated that the infection time was corrected. But it is not clear how it was corrected and based on what.

Reply: The time since HIV-1 infection date was estimated based on indicators as described in our methods section. Originally, we determined the initiation of ART in acute/chronic infection phase using estimated infection date and adjusted it using categorical variables. Following this reviewers' previous comment 4, we performed and showed in the first round of revisions a sensitivity analysis of adjusting time since infection numerically, which did not disrupt our heritability estimates. Further, please note that time of infection and within- patient evolution become less relevant for liberal thresholds, which argues for showing a variety of models (see your concern 4). Finally, please note that HIV-1 accumulates less than 1% diversity per year^{12,13}.

Also, we made the following changes in the manuscript to further clarify the adjustment for infection

time:

Line 502: "We determined the initiation of ART as occurring in acute/chronic infection using the estimated HIV-1 infection date as specified in the previous study³."

Comment Reviewer #4:

I agree that this could be better defined rather than referring to the previous published paper of the same group. This point has been addressed in the first round of review and If reviewer #1 and #2 who have expertise with virus evolution and heritability agree with this, this is OK for me.

3. As a virologist, I still don't understand how the models work. The authors should explain this to the general readers better. Are all variants equally important to the heritability? If so, how can synonymous mutations contribute? How APOBEC mediated hypermutations that is very frequent in proviral genome sequences affect the models? Can heritability be determined by some particular mutations? If so, can this be identified.

Reply: This might be a misunderstanding. In both datasets, we sequenced plasma viruses. Thus, this has nothing to do with proviruses. We further clarified this in line 399:

"5) either A. NGS sequencing of the full genome from plasma at the latest sample before the initiation of ART or B. Sanger sequencing of partial pol region obtained from plasma for genotypic resistance testing (GRT)"

Further, please note that phylogenetic models generally do not treat different variants of the HIV-1 genome differently and that our model compares if viruses with a similar genome have similar phenotypes (which is how heritability is defined).

Comment Reviewer #4:

I do agree with the authors. The paper is focusing on latest plasma sample before initiation of cART. The reviewer was probably confused due to the relationship with the reservoir size, but a qualitative assessment of the reservoir is lacking (this is the whole point of discussion indeed). It might however be confusing how the authors make conclusions based on the NGS data. This NGS data is not looking at diversity (which you would expect from NGS) as only one consensus sequencing is used for downstream analysis. We can also not talk about intact sequences as this analysis is not done on single virus level but on bulk viral population. So, the data on different variants present as a viral swarm in each patient is not taken into account. The complete method description on how the NGS is performed is also lacking in the current paper. So, the authors use the consensus sequence from the NGS data but the bioinformatic pipeline how to build this consensus sequences is lacking. Nevertheless, if one only looks at the consensus sequence, it is indeed possible to study heritability as demonstrated in this manuscript.

4. While it is understandable that the authors tried to present many different models and use different viral regions to present as many scenarios as possible. But the high variable results among them will make the readers completely puzzled about what should be the best to use.

Reply: We acknowledge the reviewers' concern. However, the aim of this paper is not to identify the "best" method to measure heritability but rather to provide the reader with a full and systematic assessment of heritability based on the available methodology. Please further note, that no single "best" heritability estimator exists¹⁴ (main point 6 of our reply in the first round of revisions), but that among the heritability estimators each method has advantages and disadvantages. Finally, we strongly believe that it is a key strength of this study that it provides a systematic assessment of heritability estimates for HIV-1 reservoir size and decay dynamics by applying different heritability estimation methods to distinct datasets (thereby showing the true uncertainty of these estimates). This systematic nature of the assessment strengthens our conclusions by showing a consistent pattern

of a substantial heritability of the HIV-1 reservoir size but weaker patterns for the HIV-1 decay slope.

Comment Reviewer #4:

Here I would propose to rely on Reviewer #1 and #2 who have expertise with virus evolution and heritability.

Point-by-point reply (round 3)

We appreciate the new reviewers' positive assessment of both our revised manuscript and our previous point-to-point replies. We would like to thank reviewer #4 for her/his profound comments and clarifications, which we addressed in this point-by-point reply. Our point-by-point reply contains reviewer #3's comments and our previous replies in blue, reviewer #4's new comments in grey, our answers in black, and changes in the manuscript in brown.

Reviewer #4 (Remarks to the Author):

Dear Mrs Schmid,
Dear Mr Rosch,

Concerning the manuscript from Wan and Bachmann entitled: Heritability of the HIV-reservoir size and decay under long-term suppressive ART. The key questions for me have indeed been formulated by Mrs Schmid:

In short, the paper has already been through two rounds of review. Reviewer #1 and #2 who have expertise with virus evolution and heritability had only minor remaining concerns that authors have addressed appropriately. However, reviewer #3 who has expertise with HIV reservoir had major remaining concerns on study design, more specifically on the use of HIV DNA as proxy for the reservoir size.

We feel that the authors and reviewer #3 are at an impasse and are seeking the opinion on this issue from an additional independent scientist. We are hoping that you will be willing to comment on the concerns raised by reviewer #3 and let us know whether in your opinion the authors have appropriately addressed these concerns and whether in your opinion the manuscript is suitable for publication in Nature Communications.

I therefore focused on the specific questions and the concerns raised by reviewer #3 and give my overall thoughts at the end of this document.

Reviewer #3 (Remarks to the Author):

This reviewer appreciates the effort that authors put into their responses to address concerns from all reviewers, especially analyzing the subtype B sequences separately to avoid the unpredictable effects of different subtypes. Unfortunately, some of my major concerns are not fully addressed.

1. The total DNA level simply cannot be used to as a reliable proxy for the reservoir size. The leading experts, Drs. Siliciano, Coffin and others, have made this very clear. The reason is that the total DNA level is way too bigger (the vast majority of them are from defective genomes) than the real reservoir size in a patient. Thus, the total DNA amount in a sample can completely overshadow the size of the reservoir. However, it is ok to use total DNA levels to measure the DNA loads in the samples.

Reply: The reviewers' concern points to an ongoing discussion in the field: While we of course agree that total HIV-1 DNA levels are higher than the HIV-1 reservoir of replication competent latently infected cells, we must disagree that total HIV-1 DNA cannot be used as a proxy for the HIV-1 reservoir. The reasons for this have been outlined in detail in the first round of our revisions and were integrated in the discussion section of the manuscript in line 367. Especially the following points should be noted:

In our previous study¹, we found that total HIV-1 DNA measured in PBMC is a robust proxy for the latent HIV-1 reservoir size after the first rapid decay following initiation of ART for several reasons: It correlates independently with time to initiation of ART, with CD4+ cell count, and with viral blips. In addition, the decay of the total HIV-1 reservoir we observed¹ was consistent with that observed in smaller studies using either viral outgrowth assays or total HIV-1 DNA (published by the leading experts mentioned by the reviewer)^{2,3}. In particular we found similar decay dynamics and determinants to those studies. Also other studies have found total HIV-1 DNA in peripheral blood mononuclear cells samples to be a sensitive, clinically relevant marker for the HIV-1 reservoir, which in contrast to the viral outgrowth assay can be measured in larger populations^{4,5}.

Furthermore, regarding the defective genomes, a recent study by Anthony Fauci's group highlighted that defective proviruses are not silent and are capable of transcribing novel unspliced forms of HIV-RNA transcripts during cART⁶. This phenomenon may further help to demonstrate the biological relevance of total HIV-1 DNA, i. e., that not just replication competent DNA genomes are of biological relevance but also defective proviruses. Such findings may potentially also be relevant for residual inflammation that is found in patients with successfully suppressed viremia.

More generally, it is important to note that so far there is no single perfect measurement for the latent HIV-1 reservoir. This is the case for many reasons: i) anatomically: one will never be able to sample all anatomical and cellular reservoirs at a given timepoint simultaneously in people living with HIV and being alive, ii) the variety of assays available (e.g. qVOA, TILDA, PCR based assays for total, integrated and non-integrated proviral DNA, for replication competent proviral DNA) all have their strength and weaknesses, iii) the very recently described, sophisticated methods are not applicable to a cohort of over 1000 participants, as they are too time- and labor intensive and/or require a high amount of cells, and thus, are only (prospectively) applicable to selected individuals^{7,8}. Such studies of a few individuals, although highly interesting and important, always bear the risk of strong selection implying that generalizability to the population level is often problematic. Thus, there is a trade-off between the potential bias directly induced by the measurement method and the bias indirectly induced by method's restrictions and limitations of the study population. In this context, we believe that our choice is a good compromise between these two conflicting requirements (and in fact we believe that it is at this time the only choice that allows to estimate viral heritability). However, we agree that at the end of the day both types of studies are needed in an iterative way to move the field forward.

Finally, a new, very recently online published paper by Papasaavas et al in "Clinical Infectious Diseases" shows a significant correlation between total proviral DNA and the intact provirus DNA assay, underlining the utility of total proviral DNA as a proxy for the viral reservoir⁹.

Comment Reviewer #4:

Total HIV DNA is related to virological failure in monotherapy studies, both for PI's and for INI's, so there is a link with total HIV DNA and functional reservoir in terms of generation of new viral particles. Reviewer 3 is right that the functional reservoir is much more complex and the field has moved towards the idea that total DNA HIV is not a reliable proxy that can be used in cure studies. Both the intactness of the sequence, the integration site and the chromosomal structure will influence potential transcriptional activity and the formation of new infectious particles. However, for large scale studies we are currently limited by the

enormous cost and the lack of high throughput assays to measure those parameters. The authors have outlined that very nicely in the rebuttal letter but only to a limited extent in the manuscript itself. Therefore this should be adapted and integration of the key points raised in the rebuttal letter should be integrated in the manuscript discussion, as this is indeed a major limitation of the study.

Reply: We would like to thank the reviewer for acknowledging our previous reply on this issue. We agree that as a major limitation of the study, this should be discussed in more detail in the manuscript. Thus, we added a new paragraph as suggested by the reviewer in line 440:

“Our study has some limitations. First, we used total HIV-1 DNA measured in peripheral blood mononuclear cells samples (PBMC) as a proxy for the latent HIV-1 reservoir. As total HIV-1 DNA levels are higher than the HIV-1 reservoir of replication competent latently infected cells, this is theoretically a limitation of our study. However, we believe our choice is the optimal compromise at this time: On the one hand, it has been shown that total HIV-1 DNA measured in PBMC samples is a sensitive, clinically relevant proxy for the HIV-1 reservoir⁹⁻¹² that also correlates well with the intact proviral DNA assay⁹. On the other hand, the very recently described, sophisticated methods (quantitative viral outgrowth assay, tat/rev induced limiting dilution assay etc) are not applicable to a cohort of over 600 longitudinally sampled participants, as they are too time- and labor intensive and/or require a high amount of cells, and thus, are only (prospectively) applicable to selected individuals^{7,8}. Such studies of a few individuals, although highly interesting and important, always bear the risk of strong selection implying that generalizability to the population level is often problematic. In contrast, total HIV-1 DNA can be determined in large populations needed for heritability studies¹⁰⁻¹². Thus, there is a trade-off between the potential bias directly induced by the measurement method and the bias indirectly induced by method’s restrictions and limitations of the study population.”

2. If the branch lengths in phylogenetic trees are used in all models, determination of infection time will be critical. Generally, the HIV-1 accumulated 1% diversity per year. Comparing the viruses from 1-year infection to those from 10-year infection will be a very different thing. The authors stated that the infection time was corrected. But it is not clear how it was corrected and based on what.

Reply: The time since HIV-1 infection date was estimated based on indicators as described in our methods section. Originally, we determined the initiation of ART in acute/chronic infection phase using estimated infection date and adjusted it using categorical variables. Following this reviewer’s previous comment 4, we performed and showed in the first round of revisions a sensitivity analysis of adjusting time since infection numerically, which did not disrupt our heritability estimates. Further, please note that time of infection and within-patient evolution become less relevant for liberal thresholds, which argues for showing a variety of models (see your concern 4). Finally, please note that HIV-1 accumulates less than 1% diversity per year^{13,14}.

Also, we made the following changes in the manuscript to further clarify the adjustment for infection time:

Line 502: “We determined the initiation of ART as occurring in acute/chronic infection using the estimated HIV-1 infection date as specified in the previous study¹.”

Comment Reviewer #4:

I agree that this could be better defined rather than referring to the previous published paper

of the same group. This point has been addressed in the first round of review and If reviewer #1 and #2 who have expertise with virus evolution and heritability agree with this, this is OK for me.

Reply: We would like to thank the reviewer for pointing it out. We elaborated the description to better define the methods used:

Line 682: “We determined the initiation of ART as occurring in acute/chronic infection using the estimated HIV-1 infection date, which was estimated using a hierarchical approach on the basis of indicators of varying reliability¹⁵.The following methods were used with decreasing priority to yield the maximal accuracy for HIV-1 infection dates possible:

1. Defined HIV-1 primary infection: Either seroconversion dates (negative and positive HIV-1 screening tests less than 1 year apart) or a diagnosis of a primary infection were available as previously described¹⁶. We used the midpoint between the dates of the negative and positive tests or, if known, the date of the primary infection as the estimated HIV-1 infection date for these individuals.
2. Defined recent HIV-1 infection: Genotypic HIV-1 drug resistance test within the first year after diagnosis revealed low HIV-1 diversity (less than 0.5% of ambiguous nucleotides) and CD4+ cell counts were >200 cells/ μ l blood at registration¹⁷⁻¹⁹, we interpreted these as recent HIV-1 infections and used the diagnosis date as an estimate for the infection date.
3. HIV-1 infection date estimates based on a back-calculation method using CD4+ cell counts and their slopes when available²⁰.
4. For the remaining individuals, no accurate dating was available. For these individuals the date of diagnosis was used as substitute for the HIV-1 infection date, which allowed us to define at least the minimum length of HIV-1 infection.

3. As a virologist, I still don't understand how the models work. The authors should explain this to the general readers better. Are all variants equally important to the heritability? If so, how can synonymous mutations contribute? How APOBEC mediated hypermutations that is very frequent in proviral genome sequences affect the models? Can heritability be determined by some particular mutations? If so, can this be identified.

Reply: This might be a misunderstanding. In both datasets, we sequenced plasma viruses. Thus, this has nothing to do with proviruses. We further clarified this in line 399:

“(5) either A. NGS sequencing of the full genome from plasma at the latest sample before the initiation of ART or B. Sanger sequencing of partial pol region obtained from plasma for genotypic resistance testing (GRT)”

Further, please note that phylogenetic models generally do not treat different variants of the HIV-1 genome differently and that our model compares if viruses with a similar genome have similar phenotypes (which is how heritability is defined).

Comment Reviewer #4:

I do agree with the authors. The paper is focusing on latest plasma sample before initiation of cART. The reviewer was probably confused due to the relationship with the reservoir size, but a qualitative assessment of the reservoir is lacking (this is the whole point of discussion indeed). It might however be confusing how the authors make conclusions based on the NGS data. This NGS data is not looking at diversity (which you would expect from NGS) as only one consensus sequencing is used for downstream analysis. We can also not talk about intact sequences as this analysis is not done on single virus level but on bulk viral population. So, the data on different variants present as a viral swarm in each patient is not taken into account.

The complete method description on how the NGS is performed is also lacking in the current paper. So, the authors use the consensus sequence from the NGS data but the bioinformatic pipeline how to build this consensus sequences is lacking.

Nevertheless, if one only looks at the consensus sequence, it is indeed possible to study heritability as demonstrated in this manuscript.

Reply: We would like to thank the reviewer for acknowledging our reply. We added the following sentences to describe the bioinformatic pipeline for building the NGS consensus sequences:

Line 556: “We used consensus sequences, which were derived from NGS data (determined as described in Supplementary Method) by the analysis pipeline V-pipe²¹. In this pipeline, NGS reads were preprocessed with PRINSEQ v0.20.4²². We then aligned the preprocessed reads to an HXB2 reference genome and generated the consensus sequences with a majority vote rule using ngshmmalign.”

4. While it is understandable that the authors tried to present many different models and use different viral regions to present as many scenarios as possible. But the high variable results among them will make the readers completely puzzled about what should be the best to use.

Reply: We acknowledge the reviewers’ concern. However, the aim of this paper is not to identify the “best” method to measure heritability but rather to provide the reader with a full and systematic assessment of heritability based on the available methodology. Please further note, that no single “best” heritability estimator exists²³ (main point 6 of our reply in the first round of revisions), but that among the heritability estimators each method has advantages and disadvantages. Finally, we strongly believe that it is a key strength of this study that it provides a systematic assessment of heritability estimates for HIV-1 reservoir size and decay dynamics by applying different heritability estimation methods to distinct datasets (thereby showing the true uncertainty of these estimates). This systematic nature of the assessment strengthens our conclusions by showing a consistent pattern of a substantial heritability of the HIV-1 reservoir size but weaker patterns for the HIV-1 decay slope.

Comment Reviewer #4:

Here I would propose to rely on Reviewer #1 and #2 who have expertise with virus evolution and heritability.

Reply: We would like to thank the reviewer for the comment. As this point has been fully discussed in previous replies to Reivewer #1 and #2, we consider this issue as resolved.

Reference:

1. Bachmann, N. *et al.* Determinants of HIV-1 reservoir size and long-term dynamics during suppressive ART. *Nat. Commun.* (2019).
2. Finzi, D. *et al.* Identification of a Reservoir for HIV-1 in Patients on Highly Active Antiretroviral Therapy. *Science* **278**, 1295 LP – 1300 (1997).
3. Siliciano, J. D. *et al.* Long-term follow-up studies confirm the stability of the latent reservoir for HIV-1 in resting CD4+ T cells. *Nat. Med.* **9**, 727–728 (2003).
4. Avettand-Fenoel, V. *et al.* Total HIV-1 DNA, a Marker of Viral Reservoir Dynamics with Clinical Implications. *Clin. Microbiol. Rev.* **29**, 859–880 (2016).
5. Anderson, E. M. & Maldarelli, F. The role of integration and clonal expansion in HIV infection: live long and prosper. *Retrovirology* **15**, 71 (2018).
6. Imamichi, H. *et al.* Defective HIV-1 proviruses produce viral proteins. *Proc. Natl. Acad. Sci.* **117**, 3704 (2020).
7. Bruner, K. M. *et al.* A quantitative approach for measuring the reservoir of latent HIV-1 proviruses. *Nature* **566**, 120–125 (2019).
8. Cohn, L. B. *et al.* Clonal CD4+ T cells in the HIV-1 latent reservoir display a distinct gene profile upon reactivation. *Nat. Med.* **24**, 604–609 (2018).
9. Papasavvas, E. *et al.* Intact HIV reservoir estimated by the intact proviral DNA assay correlates with levels of total and integrated DNA in the blood during suppressive antiretroviral therapy. *Clin. Infect. Dis.* (2020) doi:10.1093/cid/ciaa809.
10. Bachmann, N. *et al.* Determinants of HIV-1 reservoir size and long-term dynamics during suppressive ART. *Nat. Commun.* (in press).
11. Avettand-Fènoël, V. *et al.* Total HIV-1 DNA, a Marker of Viral Reservoir Dynamics with Clinical Implications. *Clin. Microbiol. Rev.* **29**, 859–880 (2016).
12. Anderson, E. M. & Maldarelli, F. The role of integration and clonal expansion in HIV infection: live long and prosper. *Retrovirology* **15**, 71 (2018).
13. Shankarappa, R. *et al.* Consistent viral evolutionary changes associated with the progression of human immunodeficiency virus type 1 infection. *J. Virol.* **73**, 10489–10502 (1999).
14. Carlisle, L. A. *et al.* Viral Diversity Based on Next-Generation Sequencing of HIV-1 Provides Precise Estimates of Infection Recency and Time Since Infection. *J. Infect. Dis.* **220**, 254–265 (2019).
15. Rusert, P. *et al.* Determinants of HIV-1 broadly neutralizing antibody induction. *Nat. Med.* **22**, 1260 (2016).
16. Rieder, P. *et al.* HIV-1 transmission after cessation of early antiretroviral therapy among men having sex with men. *AIDS* **24**, 1177 (2010).
17. Kouyos, R. D. *et al.* Ambiguous Nucleotide Calls From Population-based Sequencing of HIV-1 are a Marker for Viral Diversity and the Age of Infection. *Clin. Infect. Dis.* **52**, 532–539 (2011).
18. Ragonnet-Cronin, M. *et al.* Genetic Diversity as a Marker for Timing Infection in HIV-Infected Patients: Evaluation of a 6-Month Window and Comparison With BED. *J. Infect. Dis.* **206**, 756–764 (2012).
19. Andersson, E. *et al.* Evaluation of sequence ambiguities of the HIV-1 pol gene as a method to identify recent HIV-1 infection in transmitted drug resistance surveys. *Infect. Genet. Evol.* **18**, 125–131 (2013).
20. Taffé, P. & May, M. A joint back calculation model for the imputation of the date of HIV infection in a prevalent cohort. *Stat. Med.* **27**, 4835–4853 (2008).
21. Posada-Céspedes, S., Seifert, D., Topolsky, I., Metzner, K. J. & Beerenwinkel, N. V-pipe: a computational pipeline for assessing viral genetic diversity from high-throughput

sequencing data. *Prepr. <https://www.biorxiv.org/content/10110120200609142919v1>*
2020.06.09.142919 (2020).

22. Schmieder, R. & Edwards, R. Quality control and preprocessing of metagenomic datasets. *Bioinformatics* **27**, 863–864 (2011).
23. Fraser, C. *et al.* Virulence and Pathogenesis of HIV-1 Infection: An Evolutionary Perspective. *Science* **343**, 1243727 (2014).